# *LNMAT1* promotes lymphatic metastasis of bladder cancer via CCL2 dependent macrophage recruitment

Changhao Chen[1,2], Wang He[1,2], Jian Huang[1,2], Bo Wang[1,2], Hui Li[3], Qingqing Cai[4], Feng Su[1], Junming Bi[1,2], Hongwei Liu[1,2], Bin Zhang[5], Ning Jiang[1], Guangzheng Zhong[1], Yue Zhao[6], Wen Dong[1,2] & Tianxin Lin[1,2]

Tumor-associated macrophages (TAMs) are the most abundant inflammatory infiltrates in the tumor microenvironment and contribute to lymph node (LN) metastasis. However, the precise mechanisms of TAMs-induced LN metastasis remain largely unknown. Herein, we identify a long noncoding RNA, termed Lymph Node Metastasis Associated Transcript 1 (*LNMAT1*), which is upregulated in LN-positive bladder cancer and associated with LN metastasis and prognosis. Through gain and loss of function approaches, we find that *LNMAT1* promotes bladder cancer-associated lymphangiogenesis and lymphatic metastasis. Mechanistically, *LNMAT1* epigenetically activates CCL2 expression by recruiting hnRNPL to CCL2 promoter, which leads to increased H3K4 tri-methylation that ensures hnRNPL binding and enhances transcription. Furthermore, *LNMAT1*-induced upregulation of CCL2 recruits macrophages into the tumor, which promotes lymphatic metastasis via VEGF-C excretion. These findings provide a plausible mechanism for *LNMAT1*-modulated tumor microenvironment in lymphatic metastasis and suggest that *LNMAT1* may represent a potential therapeutic target for clinical intervention in LN-metastatic bladder cancer.

[1] Department of Urology, State key Laboratory of Oncology in South China, Sun Yat-sen Memorial Hospital, Guangdong 510120, China. [2] Guangdong Provincial Key Laboratory of Malignant Tumor Epigenetics and Gene Regulation, State Key Laboratory of Oncology in South China, Sun Yat-sen Memorial Hospital, Guangdong 510120, China. [3] Department of Biochemistry and Molecular Genetics, School of Medicine, University of Virginia, Charlottesville, VA 22908, USA. [4] Department of Medical Oncology, Sun Yat-sen University Cancer Center, State Key Laboratory of Oncology in South China, Guangzhou 510060, China. [5] Department of Hepatopancreatobiliary Surgery, Sun Yat-sen Memorial Hospital, Guangzhou 510120, China. [6] Department of Tumor Intervention, Sun Yat-sen University First Affiliated Hospital, Guangzhou 510080, China. These authors contributed equally: Changhao Chen, Wang He, Jian Huang. Correspondence and requests for materials should be addressed to T.L. (email: lintx@mail.sysu.edu.cn)

The tumor microenvironment is comprised of diverse non-malignant stromal cell types that are associated with tumor progression and metastasis[1,2], including tumor-associated macrophages (TAMs), which are the most abundant migratory hematopoietic cell type[3,4]. Numerous data from clinical and epidemiological studies have shown a strong association between TAMs density and poor prognosis in various types of human cancer[5,6], including bladder cancer[7,8]. Macrophages are heterogeneous cells that respond differently to various microenvironmental signals and thus display distinct roles in each step of cancer progression[9]. TAMs may promote tumor progression and metastasis by directly affecting the epithelial–mesenchymal transition (EMT)[10], extracellular matrix remodeling[11] and neo-angiogenesis processes[12]. Moreover, TAMs accumulation is associated with lymphangiogenesis and upregulation of the lymphogenic factor, vascular endothelial growth factor C (VEGF-C)[13,14]. However, whether targeting TAMs-mediated lymphangiogenesis could be potential approaches for the therapeutic intervention of bladder cancer lymphatic metastasis remains largely unclear.

The chemokine pattern expressed at the tumor site plays a vital role in the orientation and differentiation of macrophage phagocytes, which modulate the suitability of the tumor microenvironment for cancer progression[15,16]. Chemokine (C-C motif) ligand 2 (CCL2), which is predominantly produced by various cancer types, including bladder cancer[17], is particularly important in cancer metastasis[18]. Tumor-derived CCL2 recruits different subsets of myeloid cells, including TAMs, which contribute to cancer cell proliferation, the inflammatory microenvironment of the tumor, immune response evasion and angiogenesis[17,19]. Importantly, CCL2 targeting has shown an effective therapeutic impact in preclinical cancer models, in which neutralizing antibodies against CCL2 attenuate 96% of the tumor burden in vivo[20]. Recently, the CCL2 monoclonal antibody carlumab (CNTO 888) was tested in clinical trials[21–23]. Therefore, exploring the biological characters and molecular mechanisms underlying sustained CCL2 expression in bladder cancer may provide clinically predictive tools for effective anti-CCL2 treatments.

Long noncoding RNAs (lncRNAs), a large class of non-protein-coding transcripts that are over 200 nt in length[24], have emerged as key regulators of important biological processes involved in the development and progression of human cancers[25,26]. LncRNAs, such as HOTAIR, Xist and Drosophila roX2, can serve as molecular scaffolds, recruiting specific regulatory proteins to form unique functional complexes[27–29], which suggests that dysregulation of lncRNAs contributes to the development and progression of cancer.

Here, we identified an lncRNA LINC01296, termed Lymph Node Metastasis Associated Transcript 1 (LNMAT1), which was markedly upregulated in lymph node (LN)-metastatic bladder cancer and was significantly associated with the clinicopathological characteristics and survival of bladder cancer patients. LNMAT1 overexpression enhanced CCL2 transcription by promoting hnRNPL-mediated H3 lysine 4 trimethylation (H3K4me3) at the CCL2 promoter. Furthermore, LNMAT1-induced CCL2 expression by bladder cancer cells recruited TAMs, which participated in the pro-lymphangiogenic process via VEGF-C excretion and ultimately contributed to lymphangiogenesis and lymphatic metastasis. These findings uncover a mechanism of TAMs-mediated lymphatic metastasis and reveal the oncogenic role of LNMAT1 in bladder cancer progression.

## Results

### LNMAT1 correlates with LN-metastasis of bladder cancer. To identify critical lncRNAs that contribute to bladder cancer

progression, next-generation sequencing (NGS) was performed in five paired high-grade muscle invasive bladder cancer (MIBC) tissues and normal adjacent tissues (NATs) and in another 5 LN-positive and LN-negative bladder cancer tissues. The patient characteristics are shown in Supplementary Table 1. As shown in Fig. 1a, b, 32 and 35 lncRNAs were upregulated and down-regulated, respectively, by more than five-fold in the high-grade MIBC tissues compared with the NATs, and 35 and 25 lncRNAs were upregulated and downregulated, respectively, by more than 5-fold in the LN-positive bladder cancer tissues compared with the LN-negative samples. Three lncRNAs, including LNMAT1, CTD-2231H16.1 and BCAR4, were consistently upregulated in both the MIBC and LN-positive bladder cancer tissues (Fig. 1c). Notably, only LNMAT1 was significantly overexpressed in the further validation in a larger cohort of 266 cases of bladder cancer tissues and paired NATs, as determined by quantitative real-time PCR (qRT-PCR) analysis ($p < 0.001$, Fig. 1d and Supplementary Fig. 1a–c). LNMAT1 is located at human chromosome 14q11.2 (RefSeq accession number: MH666079, Supplementary Fig. 1d). The full-length LNMAT1 in bladder cancer cells is identified by the 5′ and 3′ rapid amplification of cDNA ends (RACE) (Supplementary Fig. 1e–f).

Furthermore, statistical analysis revealed that LNMAT1 expression was strongly correlated with pathological grade (Fig. 1e) and LN metastasis status (Fig. 1f) in a large cohort of bladder cancer patients ($n = 266$; Supplementary Table 2). The qRT-PCR analysis showed that LNMAT1 overexpression was apparent in metastatic tumor cells in the LNs compared with the paired primary tumors, suggesting that LNMAT1 might be a key component of metastatic cells (Fig. 1g). Importantly, high LNMAT1 expression was associated with poor overall survival (OS) and disease-free survival (DFS) in bladder cancer patients ($p < 0.01$) (Fig. 1h, i). Univariate and multivariate Cox proportional hazards analyses showed that LNMAT1 expression was an independent prognostic factor for OS (Supplementary Table 3) and DFS (Supplementary Table 4) in bladder cancer patients. Consistent with the results obtained by qRT-PCR, in situ hybridization (ISH) analysis showed that LNMAT1 expression was marginally detected in NATs and slightly increased in the LN-negative bladder cancer tissues but strongly upregulated in the LN-positive bladder cancer tissues (Fig. 1j). These data suggest that LNMAT1 plays vital roles in lymph node metastasis of bladder cancer.

Moreover, analyses of The Cancer Genome Atlas (TCGA) database showed that LNMAT1 was significantly overexpressed in various types of human cancer, such as bladder, prostate, kidney, colon, lung, and liver cancer (Supplementary Fig. 2a–f), and LNMAT1 overexpression correlated with LN metastasis (Supplementary Fig. 2g–h) and poor prognosis in human cancers, including thyroid carcinoma, kidney cancer, colon carcinoma, and liver cancer (Supplementary Fig. 2i–p), which further suggested that LNMAT1 may play an oncogenic role in progression and development of various human cancer types.

### LNMAT1 promotes lymphatic metastasis in vivo. To further determine the role of LNMAT1 in LN metastasis of bladder cancer, an in vivo nude mouse popliteal LN metastasis model was employed (Fig. 2a), which simulates the directional drainage and metastasis of lymph nodes of bladder cancer. Lymph drainage in the footpad is directional with the popliteal and subsequent to the external iliac nodes and the common iliac lymph nodes, which are the common sites of lymph node metastases of bladder cancer. Furthermore, the well-defined lymph drainage from footpad injections enables more sensitive and quantitative in vitro measurements of lymphatic metastases.

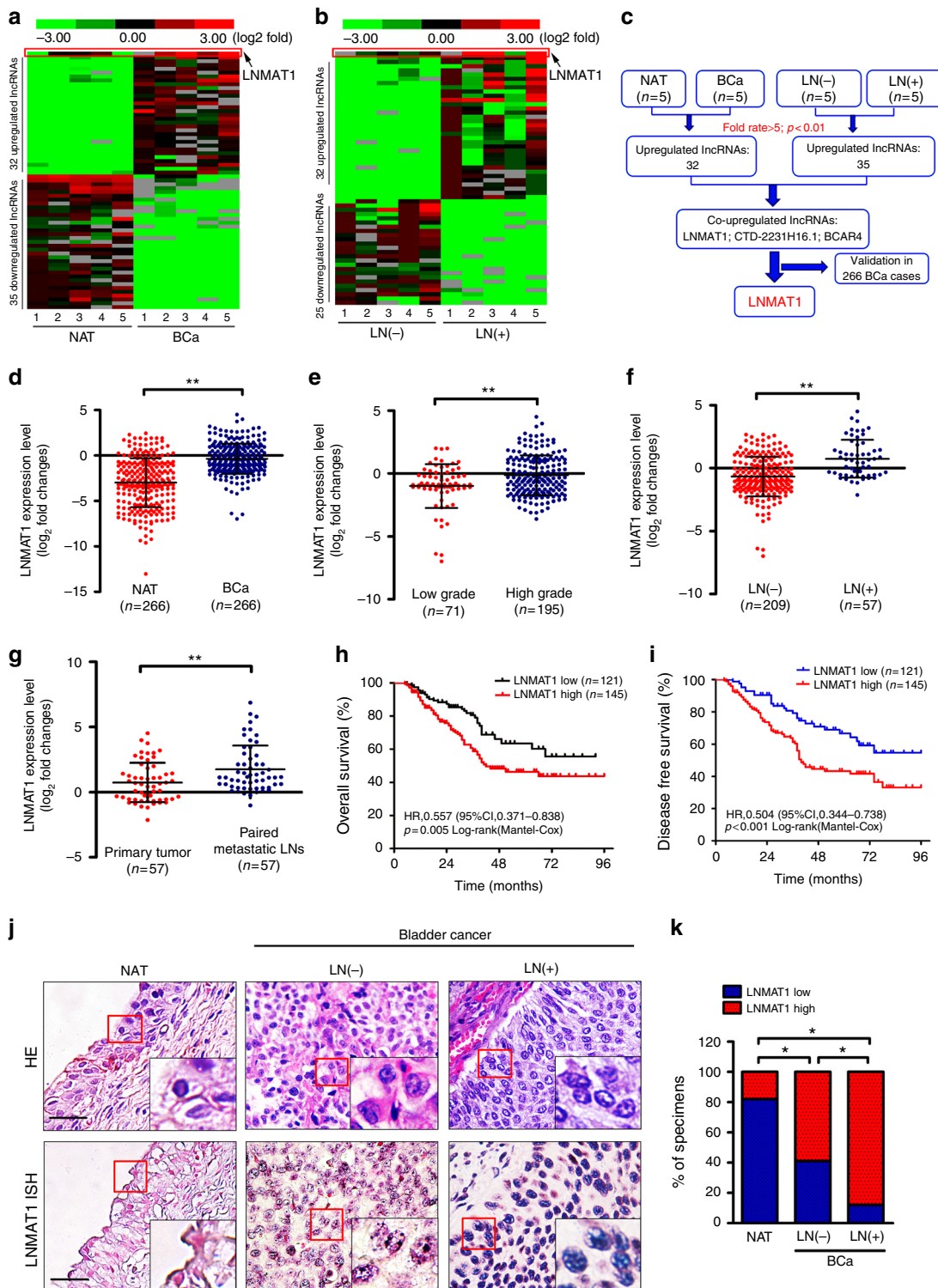

The UM-UC-3/luc and 5637/luc bladder cancer cell lines stably overexpressing *LNMAT1* or a short hairpin RNA (shRNA) targeting *LNMAT1* were inoculated into the footpads of nude mice (n = 16 per group) (Fig. 2b, c). The impact of *LNMAT1* on lymphatic metastasis was checked when the vector-control tumors reached the same size as the *LNMAT1* tumors. Strikingly, silencing *LNMAT1* notably inhibited LN metastasis. Conversely, *LNMAT1* overexpression promoted the ability of the bladder cancer cells to metastasize to the LNs, as determined by luminescence intensity and the number of metastatic LNs (Fig. 2d, e and Supplementary Fig. 3a, b). The volumes of the LNs were larger in the *LNMAT1* tumor group than in the control group, whereas the volumes of the popliteal LNs were significantly smaller in the sh-*LNMAT1* group than in the control group (Fig. 2f and Supplementary Fig. 3c). In addition, the survival times of the mice bearing *LNMAT1*-silenced tumors were longer than those of the control mice and the survival times of the *LNMAT1*-overexpressing mice were shorter than those of the

**Fig. 1** *LNMAT1* overexpression is associated with poor prognosis for bladder cancer. **a**, **b** Unsupervised hierarchical clustering of the lncRNAs differentially expressed in MIBC tissues and paired normal adjacent tissues NAT (**a**) and in LN-positive and LN-negative bladder cancer tissues (**b**). The pseudocolor represents the intensity scale of MIBC tissues vs. NAT or the LN-positive vs. LN-negative bladder cancer tissues, generated by a $\log_2$ transformation (fold changes > 5.0, $p < 0.01$). **c** Schematic representation of *LNMAT1* upregulation in MIBC tissues and LN-positive tissues. **d** qRT-PCR analysis of *LNMAT1* expression in a 266-case cohort of freshly collected human bladder cancer samples and NATs. The nonparametric Mann–Whitney *U*-test was used. **e**, **f** Correlation of *LNMAT1* expression in bladder cancer tissues ($n = 266$) assessed by qRT-PCR with pathological grade (**e**) and LN status (**f**). The nonparametric Mann–Whitney *U*-test was used. **g** Comparison of *LNMAT1* expression in primary human bladder cancer samples and paired metastatic LNs. The nonparametric Mann–Whitney *U*-test was used. **h**, **i** Kaplan–Meier curves for OS (**h**) and DFS (**i**) of bladder cancer patients with low vs. high expression of *LNMAT1*. The median *LNMAT1* expression was used as the cutoff value. **j**, **k** Representative images (**j**) and percentages (**k**) of the ISH of *LNMAT1* expression (blue) in the paraffin-embedded NAT and tumor sections of bladder cancer with or without LN metastasis ($n = 266$). Samples were counterstained with nuclear fast red. The scramble probe was used as a negative control. Statistical significance was assessed by $\chi^2$ test. Scale bars: 50 μm. The error bars represent standard deviations of three independent experiments. *$p < 0.05$ and **$p < 0.01$

control mice (Fig. 2g and Supplementary Fig. 3d). Collectively, our results suggest that *LNMAT1* overexpression contributes to the LN metastasis of bladder cancer cells in vivo.

**LNMAT1 promotes lymphangiogenesis in vivo.** Tumor-associated lymphangiogenesis is a rate-limiting step for the LN metastasis of cancer[14,30]. Since *LNMAT1* levels were significantly associated with LN metastasis, we examined whether silencing *LNMAT1* inhibited lymphangiogenesis in a nude mouse model. ISH and immunohistochemistry (IHC) analyses revealed that *LNMAT1* expression was significantly correlated with the density of microlymphatic vessels in both the intratumoral and peritumoral regions, as indicated by LYVE-1-positive microvessels (Fig. 2h, i, Supplementary Fig. 3e, f and Supplementary Fig. 4a, b), suggesting that *LNMAT1* promotes lymphangiogenesis in vivo. However, tube formation assays showed that *LNMAT1* overexpression in bladder cancer cells alone had little effect on lymphangiogenesis in vitro (Supplementary Fig. 4c), indicating that other factors might be involved in the *LNMAT1*-induced lymphangiogenesis of bladder cancer. On the other hand, we found that the macrophage density, as indicated by the macrophage marker F4/80, in the intratumoral and peritumoral regions of the *LNMAT1*-silenced tumors was significantly decreased compared with the negative control (NC) tumors in vivo (Fig. 2h, i, Supplementary Fig. 3e, f and Supplementary Fig. 4d), suggesting that macrophage might be involved in *LNMAT1*-induced lymphangiogenesis.

**LNMAT1 promotes the invasiveness of bladder cancer cells.** LN metastasis is a complex multistep process. In addition to intratumoral and peritumoral lymphangiogenesis in vivo, enhanced cell migration and invasion is essential for metastasis[31,32]. As shown in Fig. 3a–d, Transwell cell migration/invasion and wound healing assays revealed that *LNMAT1* overexpression significantly increased the migration and invasion in the UM-UC-3 and 5637 cells, whereas *LNMAT1* depletion had the opposite effect. Consistent with the in vitro results, tail vein assay data showed that *LNMAT1* overexpression strongly increased lung colonization, which was confirmed by HE staining and IHC staining with anti-luciferase antibody, compared with the control group (Fig. 3e–g). These results indicate that *LNMAT1* promotes the migration and invasiveness of bladder cancer cells.

**CCL2 is required for LNMAT1-induced LN metastasis.** To identify potential target genes of *LNMAT1*, NGS was performed and showed that multiple cytokines, such as CCL2 and CCL5, were significantly downregulated in the *LNMAT1*-silenced UM-UC-3 and 5637 cells compared with the control cells (Fig. 4a, b). Furthermore, Bio-Plex immunoassays showed that the secreted CCL2 protein was the most significantly increased cytokine in the

*LNMAT1*-transduced bladder cancer cells, but the secreted CCL2 protein levels were decreased in the *LNMAT1*-silenced cells (Fig. 4c, d), results that were further validated by qRT-PCR and ELISA (Fig. 4e–h). Collectively, our results suggest that *LNMAT1* induce CCL2 expression in bladder cancer.

Next, we examined whether the blocking CCL2 signaling using a CCL2-neutralizing antibody inhibited *LNMAT1*-induced LN metastasis. Consistent with previous reports[33,34], CCL2 inhibition significantly inhibited the migratory capability of bladder cancer cells (Supplementary Fig. 5a, b). Furthermore, the effects of CCL2 inhibition with a CCL2-neutralizing antibody or RNA interference (RNAi) technology on *LNMAT1*-induced LN metastasis were determined in vivo. As shown in Fig. 4i and Supplementary Fig. 6a, b, treatment with the CCL2-neutralizing antibody significantly decreased the ability of UM-UC-3 cells to metastasize to the LNs, as determined by luminescence and the number of metastatic LNs. In addition, CCL2-neutralizing antibody notably reduced the *LNMAT1*-transduced tumor burden in the LNs, which led to prolonged survival times of the tumor-bearing nude mice (Fig. 4j, k), suggesting that ablation of CCL2 could inhibit *LNMAT1*-induced LN metastasis. Meanwhile, we found that treatment of cells with anti-CCL2 antibody only partially inhibited the LN-metastatic capability of the control bladder cancer cells (the ratio of metastatic LNs from 37.50% reduced to 25.00%) but strongly inhibited the LN-metastatic capability of *LNMAT1*-overexpressing cells (the ratio of metastatic LNs from 93.75% reduced to 18.75%) (Supplementary Fig. 6c). Consistently, silencing CCL2 also only moderately reduced the LN-metastatic capability of vector-control bladder cancer cells (from 31.25% reduced to 18.75%) but remarkably inhibited the LN-metastatic capability of *LNMAT1*-overexpressing cells (from 81.25 to 12.50%; Fig. 4l, m and Supplementary Fig. 6d). These results provided further evidence that tumor-cell-derived CCL2 plays a vital role in *LNMAT1*-induced lymphatic metastasis.

**LNMAT1 forms a DNA-RNA triplex with the CCL2 promoter.** To further identify the mechanisms underlying *LNMAT1*-induced CCL2 expression, we performed fluorescence in situ hybridization (FISH) and subcellular fractionation assays and showed that *LNMAT1* was predominantly localized to the nucleus (Fig. 5a, b and Supplementary Fig. 7a, b), which suggests that *LNMAT1* exerts its biological function in the nucleus. To further explore the molecular mechanisms through which *LNMAT1* stimulates CCL2 upregulation, a series of CCL2-luc promoter constructs, ranging from −2000 nt to +121 nt relative to the transcription start site, were cloned. As shown in Fig. 5c and Supplementary Fig. 7c, the promoter luciferase assay revealed an obvious increase in the transcriptional activity of the construct from −200 to +121 bp rather than from +1 to +121 bp. Moreover, a chromatin isolation by RNA purification (ChIRP) assay, which determines the exact locations of lncRNA binding sites on

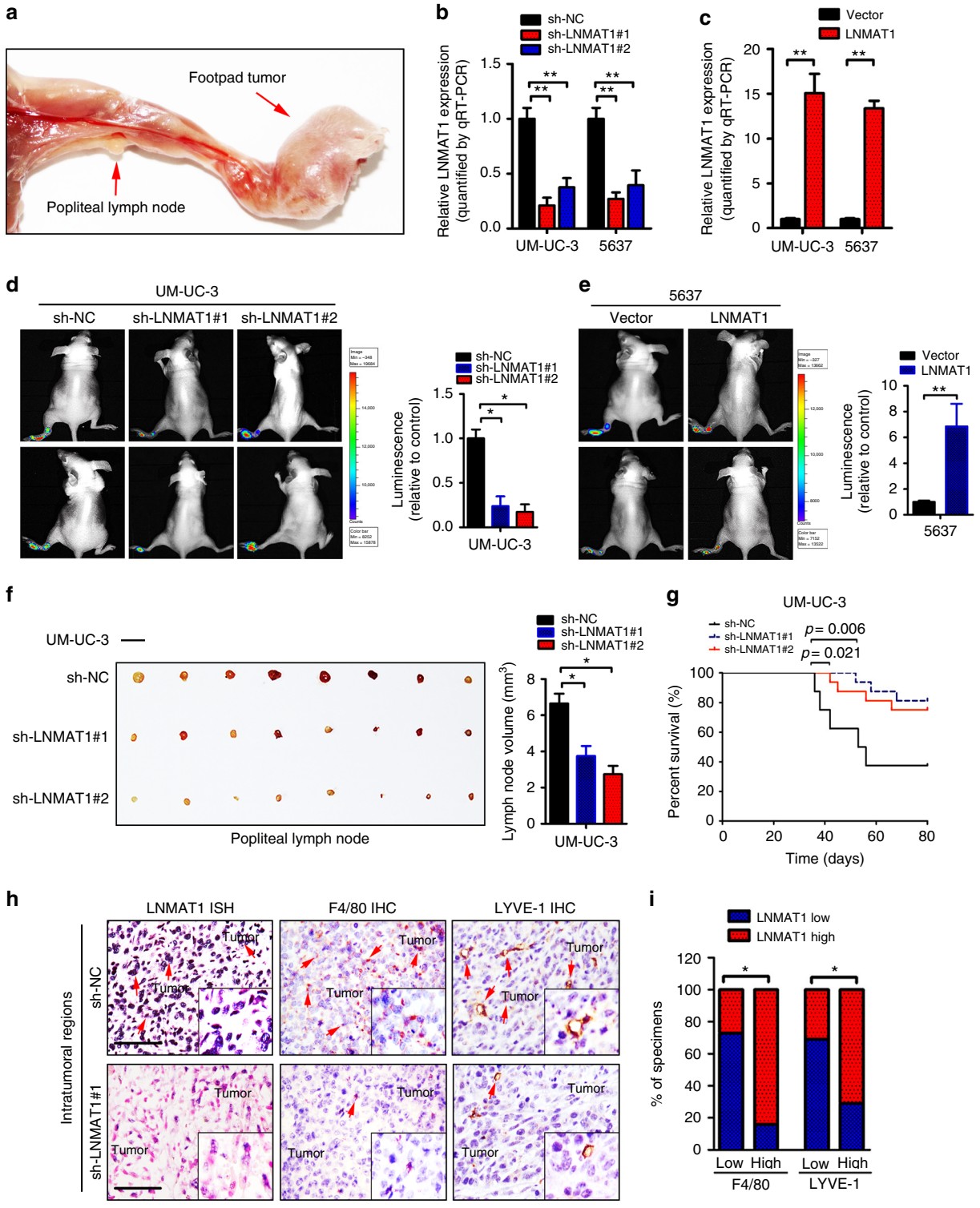

the chromatin[35], revealed that *LNMAT1* bound to −10 to −118 bp (referred to as CCL2-S2; Fig. 5d–f). Furthermore, analysis with LongTarget, a web-based tool designed to predict lncRNA-DNA binding motifs and binding sites via Hoogsteen and reverse Hoogsteen interactions[36], identified six potential triplex-forming oligonucleotides (TFOs) within *LNMAT1* and the corresponding triplex target sites (TTS) in the CCL2 promoter for possible pairing (Supplementary Table 5). Next, we synthesized the predicted TFO labeled with 5-carboxy tetramethyl-rhodamine (TAMRA) and TTS labeled with fluorescein amidite (FAM) and

performed fluorescence resonance energy transfer (FRET) analysis. FRET showed an obvious increase in the TAMRA fluorescence intensity at 570–580 nm and a decrease in 6-carboxyfluorescein (6-FAM) at 520 nm in the *LNMAT1* (462–479 nt)/CCL2 TTS4 (−60 to −43 bp) group compared with that of the control ssRNA/ CCL2 TTS group (Fig. 5g, h), which was similar to the FENDRR/PITX2-positive control group (Fig. 5i). These data indicate that FRET took place between the fluorescein donor and the rhodamine acceptor, which directly correlats with the formation of triple helices[37]. Circular dichroism

**Fig. 2** *LNMAT1* overexpression promotes LN metastasis in vivo. **a** Representative images of the nude mouse model of popliteal LN metastasis. The indicated UM-UC-3 cells were injected into the footpads of the nude mice, and the popliteal LNs were enucleated and analyzed. **b, c** qRT-PCR analysis of *LNMAT1* expression in *LNMAT1*-transduced, *LNMAT1*-silenced and control cell as indicated. Statistical significance was assessed using one-way analyses of variance (ANOVA) followed by Dunnett's tests for multiple comparison and two-tailed *t*-tests. **d, e** Representative images of bioluminescence (popliteal LNs) and histogram analysis of popliteal LN metastasis in the indicated cells ($n = 16$ per group). Statistical significance was assessed using one-way analyses of variance (ANOVA) followed by Dunnett's tests and two-tailed *t*-tests. **f** Representative images of enucleated popliteal LNs and histogram analysis of the LN volume in the indicated cells. Statistical significance was assessed using one-way analyses of variance (ANOVA) followed by Dunnett's tests and two-tailed *t*-tests. ($n = 16$, Scale bar: 5 mm). **g** Kaplan–Meier (Mantal-Cox) test of the mice ($n = 16$) that were inoculated in the indicated cells. **h, i** Representative images and percentages of mice tissues with high or low levels of F4/80-positive cells and LYVE-1-positive cells in the intratumoral tissues with different *LNMAT1* expression levels. *LNMAT1* expression levels were quantified by ISH, macrophage density was quantified by IHC using anti-F4/80 antibody and microlymphatic vessel density was quantified by IHC using the anti-LYVE-1 antibody. Two representative cases are shown. Statistical significance was assessed by $\chi^2$ test. Scale bars: 100 μm. The error bars represent standard deviations of three independent experiments. *$p < 0.05$ and **$p < 0.01$

(CD) spectroscopy[38] showed that the *LNMAT1* (462–479 nt)/CCL2 TTS4 (−60 to −43 bp) group had a distinct peak at 270–280 nm and a strong negative peak at 210 nm (Fig. 5j, k), which was similar to the FENDRR/PITX2-positive control group (Fig. 5l), suggesting that *LNMAT1* directly formed triplexes with the CCL2 promoter sequence in vitro. Together, these data suggest that *LNMAT1* regulates CCL2 transcription through direct triplex formation with the promoter sequence.

**LNMAT1 directly interacts with hnRNPL**. We subsequently performed an RNA pull down assay using in vitro transcribed biotinylated *LNMAT1* and an antisense control to identify *LNMAT1*-interacting proteins in UM-UC-3 cells. An obvious band at between 50 and 75 kDa was specifically enriched in the *LNMAT1* pull down proteins (Fig. 5m). hnRNPL was identified as the most abundant *LNMAT1*-interacting proteins via mass spectrometry (MS) (Supplementary Fig. 8). To validate the physical interaction between *LNMAT1* and hnRNPL, we performed RNA pull down followed by western blot using hnRNPL antibody. The results showed that *LNMAT1* specifically interacted with hnRNPL but not with hnRNPQ or hnRNPA1 (Fig. 5n, o). Consistently, RIP assays using nuclear extract or RNA pull down assay with purified recombinant hnRNPL protein demonstrated that *LNMAT1* directly interacted with hnRNPL (Fig. 5p and Supplementary Fig. 9a). Serial deletion analysis demonstrated that 5′-terminal region of the *LNMAT1* (350–550 nt) were required for direct interaction with hnRNPL (Fig. 6a), which was further confirmed by RNA pull down with truncated *LNMAT1* (Fig. 6b, c). Moreover, overexpression of the truncated *LNMAT1* (350–550 nt) in *LNMAT1*-silenced bladder cancer cells restored the CCL2 upregulation function of *LNMAT1* (Fig. 6d–f). These data suggest that, in bladder cancer cells, *LNMAT1* regulates CCL2 expression through direct interaction with hnRNPL.

**LNMAT1 promotes H3K4 trimethylation at the CCL2 promoter**. Next, the regulatory effect of hnRNPL on CCL2 expression was examined with ELISA and qRT-PCR assays, which showed that CCL2 expression was decreased in hnRNPL-silenced bladder cancer cells (Fig. 6g, h; Supplementary Fig. 9b, c). The luciferase activity of the CCL2 promoter was increased after the overexpression of *LNMAT1* both not mutant *LNMAT1*. Conversely, *LNMAT1* overexpression failed to activate transcriptional activity of CCL2 promoter which was mutated in the binding region of hnRNPL (Fig. 6i, j; Supplementary Fig. 9d-e), suggesting that the −118 to −10 bp region of the CCL2 promoter (CCL2-S2) was critical for *LNMAT1*/hnRNPL-induced CCL2 transactivation.

hnRNPL epigenetically regulates target gene expression by association with methylated H3K4 to catalyze H3K4me3[27]. To further confirm that *LNMAT1* activated CCL2 expression by

interacting with hnRNPL and mediating H3K4 trimethylation, Chromatin immunoprecipitation (ChIP) analysis was performed and showed that *LNMAT1* overexpression, but not mutated *LNMAT1* at 350–550 nt, dramatically enhanced hnRNPL occupancy at CCL2 promoter and increased H3K4me3 of the promoters of CCL2 (Fig. 6k, l; Supplementary Fig. 9f, g), whereas *LNMAT1* silencing drastically decreased hnRNPL occupancy and H3K4me3 of the CCL2 promoter (Fig. 6m, n; Supplementary Fig. 9h, i). Moreover, silencing hnRNPL attenuated the increased effect of *LNMAT1* on the transcriptional activation of CCL2 (Fig. 6o, p; Supplementary Fig. 9j), whereas hnRNPL overexpression partly restored the expression of CCL2 after *LNMAT1* silencing. (Fig. 6q, r; Supplementary Fig. 9k). Taken together, these data suggest that *LNMAT1* regulates CCL2 expression through hnRNPL mediated H3K4 methylation.

**CCL2-activated macrophages induce lymphangiogenesis**. Consistent with the results obtained from the mouse model, we found that *LNMAT1* expression was positively associated with the macrophage infiltration intensities of the intratumoral and peritumoral regions of bladder cancer, as indicated by the macrophage marker CD68$^+$ ($p < 0.05$; Fig. 7a, b and Supplementary Fig. 10a, b). These results suggest a potential link between *LNMAT1*-induced TAMs infiltration and lymphangiogenesis in bladder cancer. Therefore, we further examined whether *LNMAT1*-overexpressing bladder cancer cells have an impact on TAMs and tumor-induced lymphangiogenesis. We cultured fresh isolated human monocytes from healthy volunteer donors. The macrophages treated with conditioned medium (CM) from *LNMAT1*-transduced bladder cancer cells, but not with CM from the control group, displayed a stretched and elongated morphology and exhibited a CD206 high /HLA-DR low phenotype (Fig. 7c and Supplementary Fig. 10c), indicating functional activation of the macrophages. Moreover, flow cytometry analysis was performed to further assess the expression of the specific M2 markers (CD206 and CD163) and M1 markers (HLA-DR and CD86). As shown in Fig. 7c and Supplementary Fig. 11a, c, treatment with the CM derived from *LNMAT1* cells significantly increased the expression of specific M2 marker (CD206 and CD163) but decreased the expression of M1 markers (HLA-DR and CD86) compared with similar treatment in the vector cells, which provides further evidence that *LNMAT1* overexpression in cancer cells induces TAMs activation. Importantly, the addition of the CCL2-neutralizing antibody significantly suppressed the expression levels of *LNMAT1*-induced macrophages in vivo, indicating that CCL2 was necessary for *LNMAT1*-induced TAMs activation (Fig. 7d, e).

As shown in Supplementary Fig. 12a, b, treatment with CM derived from TAMs sorted from murine *LNMAT1* tumors significantly increased lymphatic capillary formation compared

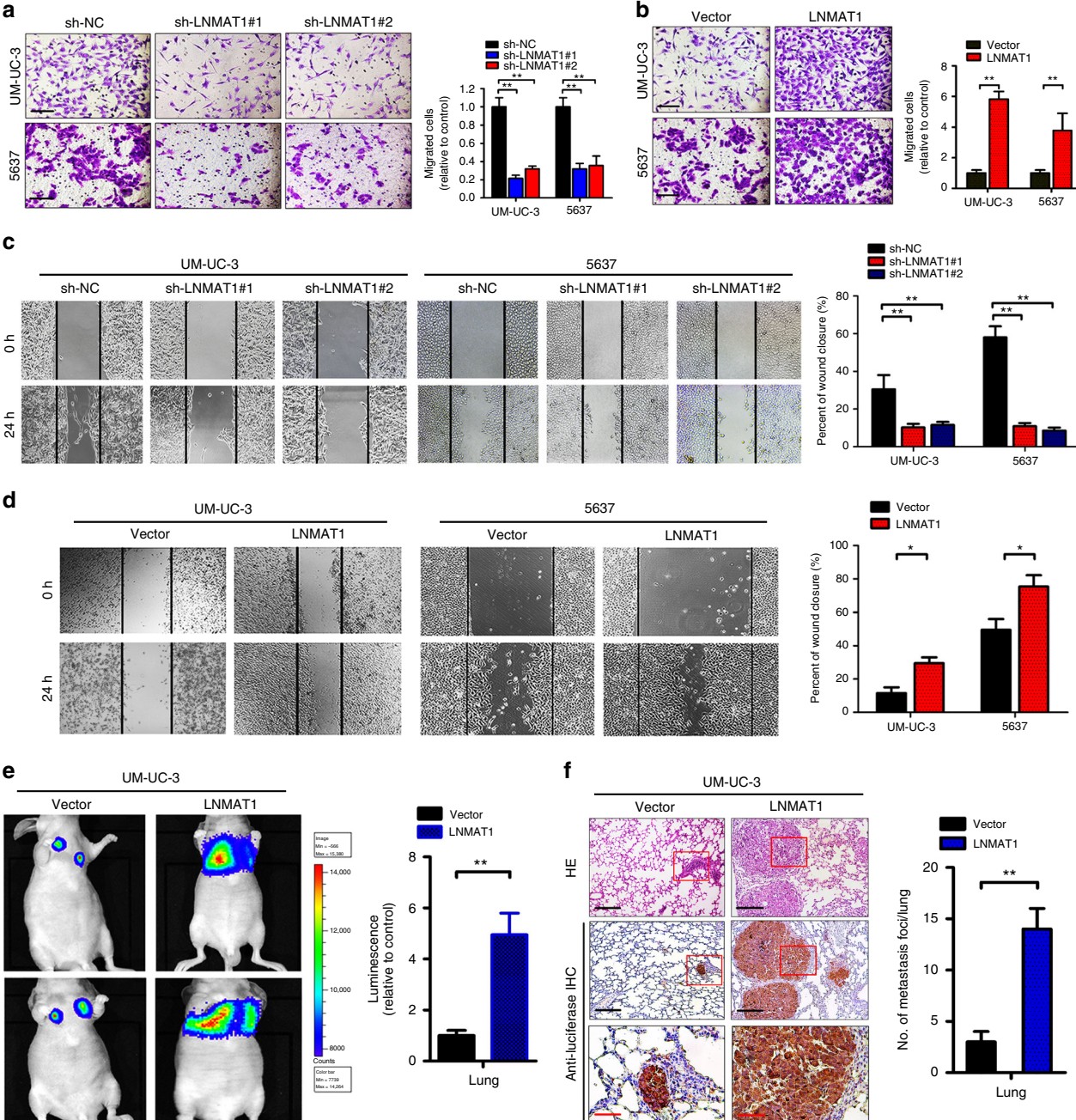

**Fig. 3** *LNMAT1* overexpression promotes bladder cancer cells invasion. **a, b** Representative images of Transwell assay using UM-UC-3 and 5637 cells showing cell motility after knockdown (**a**) or overexpression (**b**) of *LNMAT1* and a histogram analysis of migrated cell counts are shown. Scale bars: 100 μm. Statistical significance was assessed using two-tailed *t*-tests and one-way analyses of variance (ANOVA) followed by Dunnett's tests for multiple comparison. **c, d** Representative images of wound healing assay using UM-UC-3 and 5637 cells (left panels) showing cell motility after knockdown (**c**) or overexpression (**d**) of *LNMAT1* and a histogram analysis of cell migration distance are shown (right panels). Statistical significance was assessed using two-tailed *t*-tests and one-way analyses of variance (ANOVA) followed by Dunnett's tests for multiple comparison. **e–g** Representative images of lung colonization by UM-UC-3 cells injected into the tail veins of nude mice, histogram analysis of luminescence and the number of metastatic foci representing lung metastasis measured on day 60 (*n* = 16). Scale bars: black, 200 μm; red, 50 μm. Statistical significance was assessed using two-tailed *t*-tests and one-way analyses of variance (ANOVA) followed by Dunnett's tests for multiple comparison. The error bars represent standard deviations of three independent experiments. *$p < 0.05$ and **$p < 0.01$.

with CM derived from TAMs sorted from murine vector tumors, which further supports the notion that *LNMAT1* overexpression in cancer cells promotes TAMs-dependent lymphangiogenesis. In addition, we examined the effect of *LNMAT1* overexpression on the invasion of cancer cells through a lymphatic endothelium monolayer in vitro. As shown in Supplementary Fig. 12c,

overexpression of *LNMAT1* significantly promoted the migratory and invasive capability of bladder cancer cells through lymphatic endothelium monolayer, suggesting that *LNMAT1* may also increase the invasiveness of cancer cells. Therefore, our results show that *LNMAT1*-induced CCL2 might play dual roles in the lymphatic metastasis of bladder cancer cells.

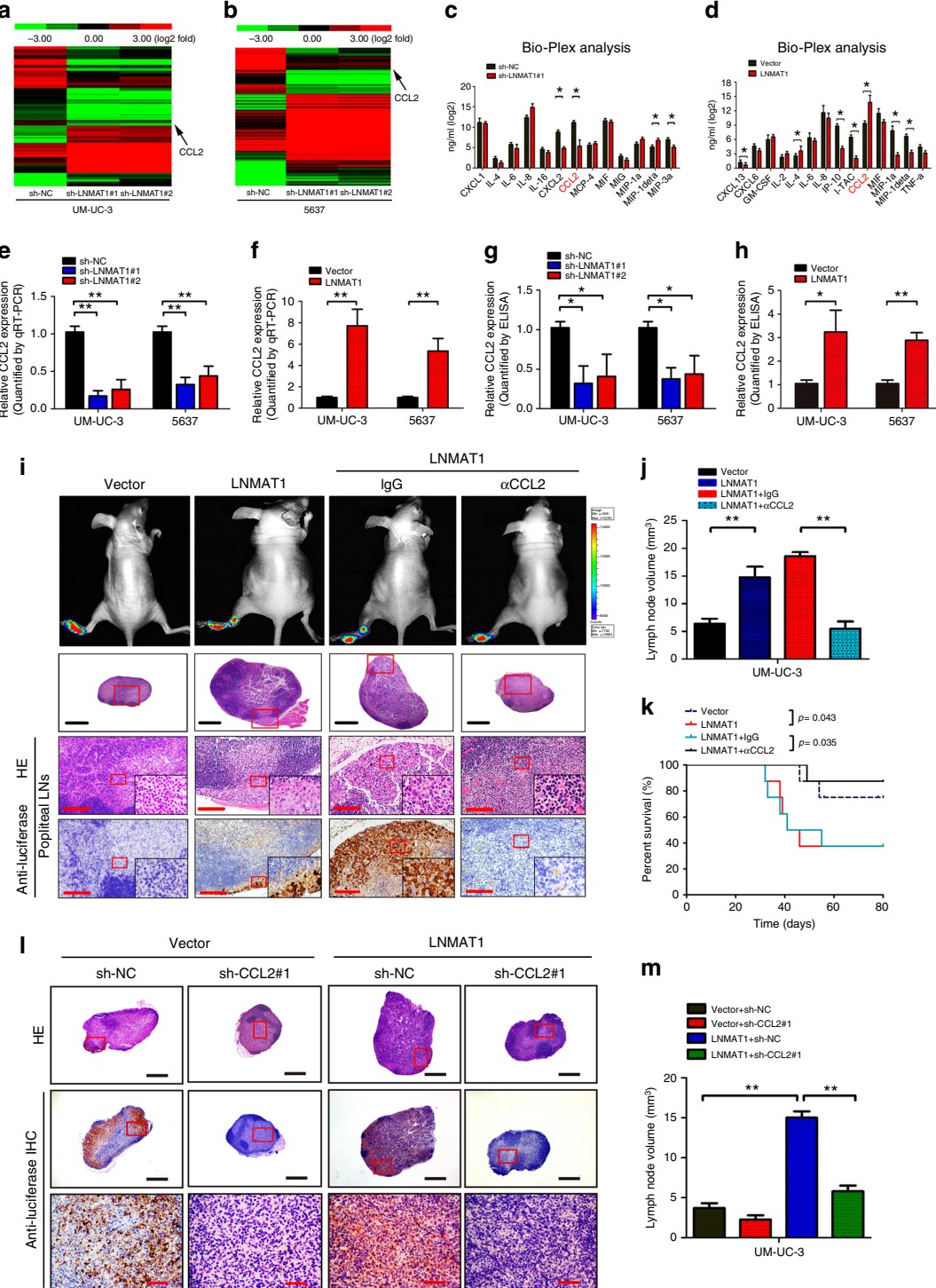

**Fig. 4** CCL2 is required for *LNMAT1*-induced LN metastasis. **a**, **b** Heatmap representing unsupervised hierarchical clustering of genes regulated by *LNMAT1* based on next-generation sequencing (NGS) analysis. Rows represent probe sets, and columns represent samples treated as indicated. Green, downregulation; red, upregulation. CCL2 is indicated by black arrows. **c**, **d** Bio-Plex multiplex immunoassay analysis of protein expression of 40 cytokines and chemokines regulated by *LNMAT1*. Two-tailed *t*-tests was used. **e**, **f** qRT-PCR of CCL2 expression in *LNMAT1*-transduced, *LNMAT1*-silenced and control cell as indicated. Statistical significance was assessed using one-way ANOVA followed by Dunnett's tests for multiple comparison and two-tailed *t*-tests. **g**, **h** ELISA of CCL2 expression in the indicated cells. Statistical significance was assessed using one-way ANOVA followed by Dunnett's tests for multiple comparison and two-tailed *t*-tests. **i** Representative images of popliteal LNs, enucleated LNs and IHC staining with anti-luciferase antibody in the indicated mice group (*n* = 16 per group). Scale bars: black, 500 μm; red, 50 μm. **j**, **k** The ratios of the metastatic to volume quantification (**j**) and Kaplan–Meier survival analysis (**k**) for the indicated group (*n* = 16 per group). Two-tailed *t*-tests was used. **l** Representative images of HE and IHC staining confirming the LN status (*n* = 16). Scale bars: black, 500 μm; red, 100 μm. **m** Volume quantification of popliteal LN metastasis after shRNA-mediated depletion of CCL2. One-way ANOVA followed by Dunnett's tests for multiple comparison was used. The error bars represent standard deviations of three independent experiments. *\*p* < 0.05 and *\*\*p* < 0.01

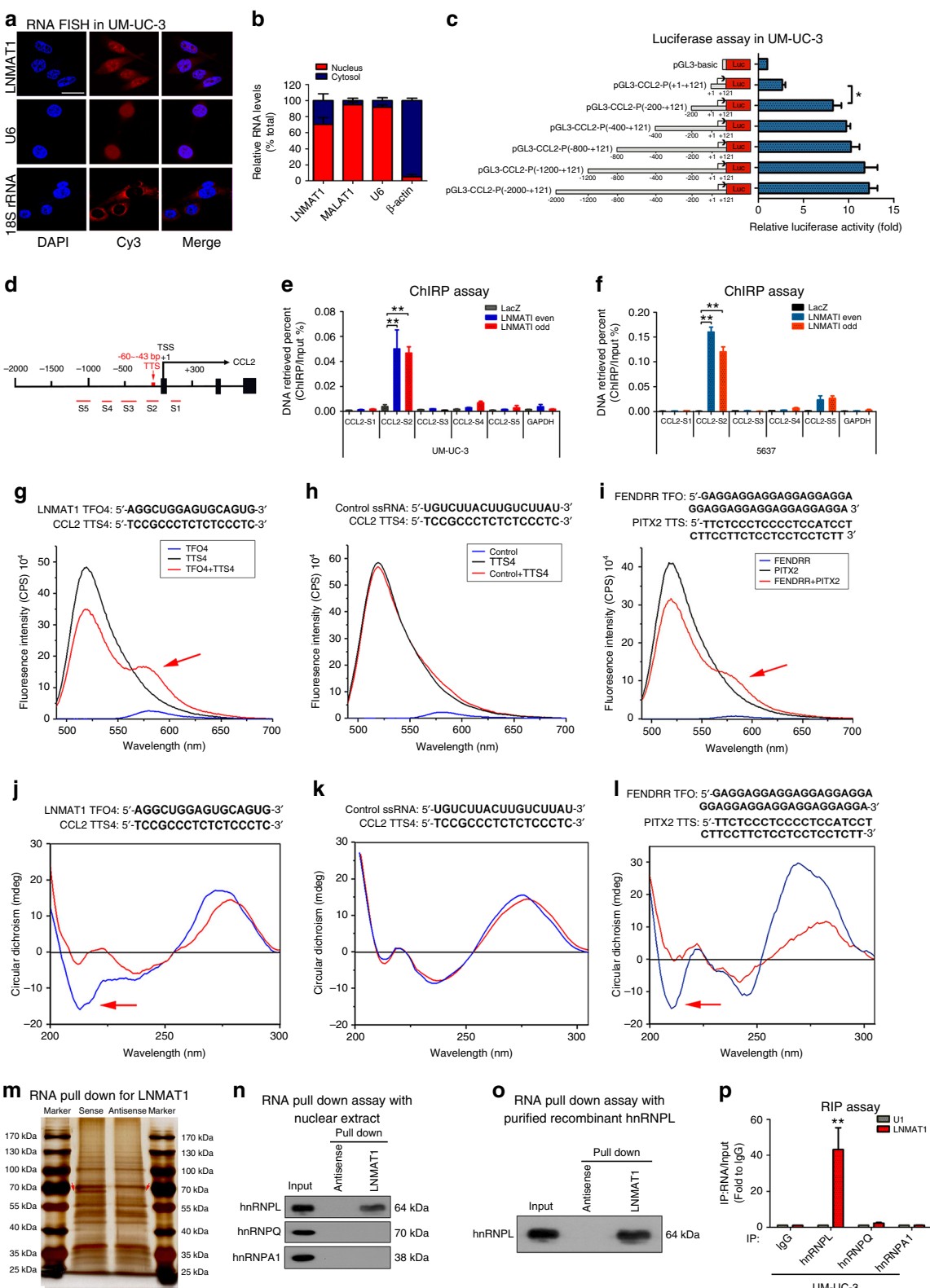

VEGF-C, a lymphangiogenic growth factor, plays key roles in the LN metastasis of multiple malignancies, including bladder cancer[39,40]. Therefore, we examined whether VEGF-C is involved in LNMAT1-induced lymphangiogenesis and whether LNMAT1-induced CCL2 excretion by bladder cancer cells could induce VEGF-C expression in TAMs. ELISA and qRT-PCR data showed that VEGF-C expression was significantly increased in CD206+/ CD163+ TAM-like macrophages induced by the addition of recombinant CCL2 (rCCL2) or CM from LNMAT1-transduced cells (Fig. 8a, b), suggesting that TAMs induced by LNMAT1-upregulated CCL2 participate in the pro-lymphangiogenic process by producing VEGF-C. Furthermore, the addition of the CCL2-neutralizing antibody carlumab (CNTO 888) to the medium collected from the LNMAT1-transduced bladder cancer

**Fig. 5** *LNMAT1* forms a triplex with the CCL2 promoter and interacts with hnRNPL. **a** FISH analysis of the subcellular distribution of *LNMAT1* in UM-UC-3 cells. Scale bar: 10 μm. **b** Nuclear fractionation analyses and qRT-PCR analyses of *LNMAT1* expression in the nucleus and cytoplasm. **c** Transcriptional activity of the CCL2 promoter was evaluated using sequential deletions and by examining the CCL2 promoter linked to Renilla luciferase activity. Two-tailed *t*-tests was used. **d** Schematic presentation of the potential *LNMAT1* binding sites in the CCL2 promoter. **e**, **f** ChIRP analysis of *LNMAT1*-associated chromatin in UM-UC-3 cells. Retrieved chromatin was quantified by qRT-PCR. Statistical significance was assessed using one-way ANOVA followed by Dunnett's tests for multiple comparison. **g–i** Fluorescence resonance energy transfer (FRET) of a 5:1 mixture of TFO in *LNMAT1* with TTS in the CCL2 promoter sequences is shown in red. The individual TFO is blue and TTS is shown in black (**g**, **h**). FENDRR/PITX2 was used as the positive control (**i**). **j–l** Circular dichroism (CD) spectroscopy of a 1:1 mixture of TFO in *LNMAT1* with TTS in the CCL2 promoter sequences is shown in blue. The sum of individual TFO and TTS is shown in red (**j**, **k**). FENDRR/PITX2 was used as the positive control (**l**). **m** An RNA pull-down assay was performed using *LNMAT1* sense and antisense RNAs in UM-UC-3 cells, followed by silver staining. A red arrow indicates hnRNPL. **n**, **o** The interaction between *LNMAT1* and hnRNPL was confirmed by RNA pull-down and western blotting with nuclear extract or purified recombinant hnRNPL. **p** RNA immunoprecipitation (RIP) analysis using the anti-hnRNPL antibody revealed that *LNMAT1* interacted with endogenous hnRNPL in UM-UC-3. U1 was used as the negative control. Statistical significance was assessed using Two-tailed *t*-tests was used. The error bars represent standard deviations of three independent experiments. *$p < 0.05$ and **$p < 0.01$

cells or the addition of the VEGF-C-neutralizing antibody to the medium collected from the TAMs significantly reduced HLECs tube formation and motility, indicating that blocking CCL2/TAMs/VEGF-C signaling inhibits *LNMAT1*-mediated lymphangiogenesis in bladder cancer (Fig. 8c–e and supplementary Fig. 13a–c). Moreover, blocking the VEGF-C signal using a VEGF-C neutralizing antibody (pV1006R-r) significantly reduced the *LNMAT1*-transduced tumor burden in the lymph nodes (Fig. 8f, g), suggesting that blocking VEGF-C signaling abrogated the *LNMAT1*-induced lymphatic metastasis in vivo. These results suggest that macrophages activated by *LNMAT1*-induced CCL2 contribute to lymphangiogenesis and LN metastasis by regulating VEGF-C.

**Clinical relevance of LNMAT1-induced upregulation of CCL2.**
Finally, we examined whether the *LNMAT1*/CCL2/lymphangiogenesis axis identified in vivo was clinically relevant to bladder cancer. We found that CCL2 was markedly overexpressed in bladder cancer tissues compared with the paired NATs (Supplementary Fig. 13d) and that CCL2 levels were positively correlated with LN metastasis ($p < 0.01$) (Fig. 9a and Supplementary Table 2). Moreover, high CCL2 expression was associated with poor OS and DFS in 266 bladder cancer patients (Fig. 9b, c). Consistently, TCGA data analysis showed that the elevated expression of CCL2 was correlated with LN metastasis (Fig. 9d), higher tumor grade (Fig. 9e) and poorer OS of bladder cancer (Fig. 9f). Furthermore, statistical analysis revealed that *LNMAT1* levels were positively correlated with CCL2 expression levels in bladder cancer specimens ($r = 0.647$, $p < 0.001$) (Fig. 9g–i). These results suggest that *LNMAT1*-induced CCL2 modulates the bladder cancer microenvironment by regulating TAMs infiltration and VEGF-C upregulation, ultimately resulting in lymphangiogenesis and lymphatic metastasis.

**Discussion**
LN metastasis confers a poor prognosis on bladder cancer patients and currently has limited treatment options in the clinic[41,42]. Thus, investigations of the molecular mechanisms underlying LN metastasis and the identification of novel, promising targets are urgently needed for prevention and therapy. Tumor microenvironment-induced VEGF-C plays a crucial role in lymphangiogenesis, which is a rate-limiting step for the LN metastasis of cancer[13,14]. However, the precise mechanism is largely unknown. Herein, we identified an lncRNA *LNMAT1*, which modulates the tumor microenvironment and thus plays an important role in the lymphatic metastasis of bladder cancer. *LNMAT1* activated CCL2 expression by recruiting hnRNPL to the CCL2 promoter, which led to increased H3K4me3 that ensured

hnRNPL binding and enhanced transcription. Mechanistically, this activation was achieved by the direct association of *LNMAT1* with DNA upstream of the transcription start site of CCL2, which forms a DNA-RNA triplex that anchors the *LNMAT1* and its associated effector proteins to the CCL2 promoter. Moreover, *LNMAT1*-induced CCL2 modulated the tumor microenvironment through TAMs infiltration and VEGF-C upregulation in bladder cancer tissues, ultimately resulting in lymphangiogenesis and lymphatic metastasis. Importantly, the inhibition of TAMs infiltration using a CCL2-neutralizing antibody dramatically reduced *LNMAT1*-induced LN metastasis in vivo. These findings provide the mechanistic and translational insight into the lncRNA-mediated modulation of the tumor microenvironment that promotes the lymphatic metastasis of bladder cancer.

TAMs have been studied extensively for their relationship with tumor cells and their multifaceted functions in the tumor microenvironment[3,19]. Evidence from recent studies showed that the accumulation of TAMs produces a host of growth factors that affect angiogenesis and tumor cell proliferation in human cancers[11,14,17]. Notably, previous studies have shown that the co-culture of tumor cells with macrophages promotes collagen degradation, highlighting a symbiotic relationship between the two cell types[43,44]. However, whether TAMs contribute to lymphatic metastasis remains unclear. Here, we demonstrated that TAMs density was positively associated with lymphatic metastasis in a clinically relevant model and nude mouse popliteal LN metastasis model. We further showed that increased TAMs infiltration, which produced higher levels of VEGF-C that promotes lymphangiogenesis and the formation of lymphatic metastasis, was induced by CCL2 secreted by bladder cancer cells. Therefore, our results provide the insights into the mechanisms underlying tumor microenvironment-mediated LN metastasis.

Another important finding in our study is that *LNMAT1* epigenetically upregulates CCL2 expression. CCL2 is highly expressed and secreted from cancer cells and contributes to cancer progression[17,45]. Additionally, CCL2 modulates the tumor microenvironment by promoting the infiltration of TAMs[46], which originate from circulating monocytes in the tumor-associated vasculature[47]. More recently, the CCL2 monoclonal antibody carlumab (CNTO 888) was examined in several phase I and II clinical trials for the treatment of solid tumors[22,23,48] (NCT Number: NCT00992186; NCT01204996; NCT00537368). Thus, exploring the precise molecular mechanism underlying sustained CCL2 expression in bladder cancer would provide a potential predictor of effective anti-CCL2 treatment. In this study, we demonstrated that *LNMAT1* epigenetically upregulated CCL2 expression in bladder cancer cells by interacting with hnRNPL to promote H3K4me3 of the CCL2 promoter. The CCL2-neutralizing antibody dramatically decreased the burden of the

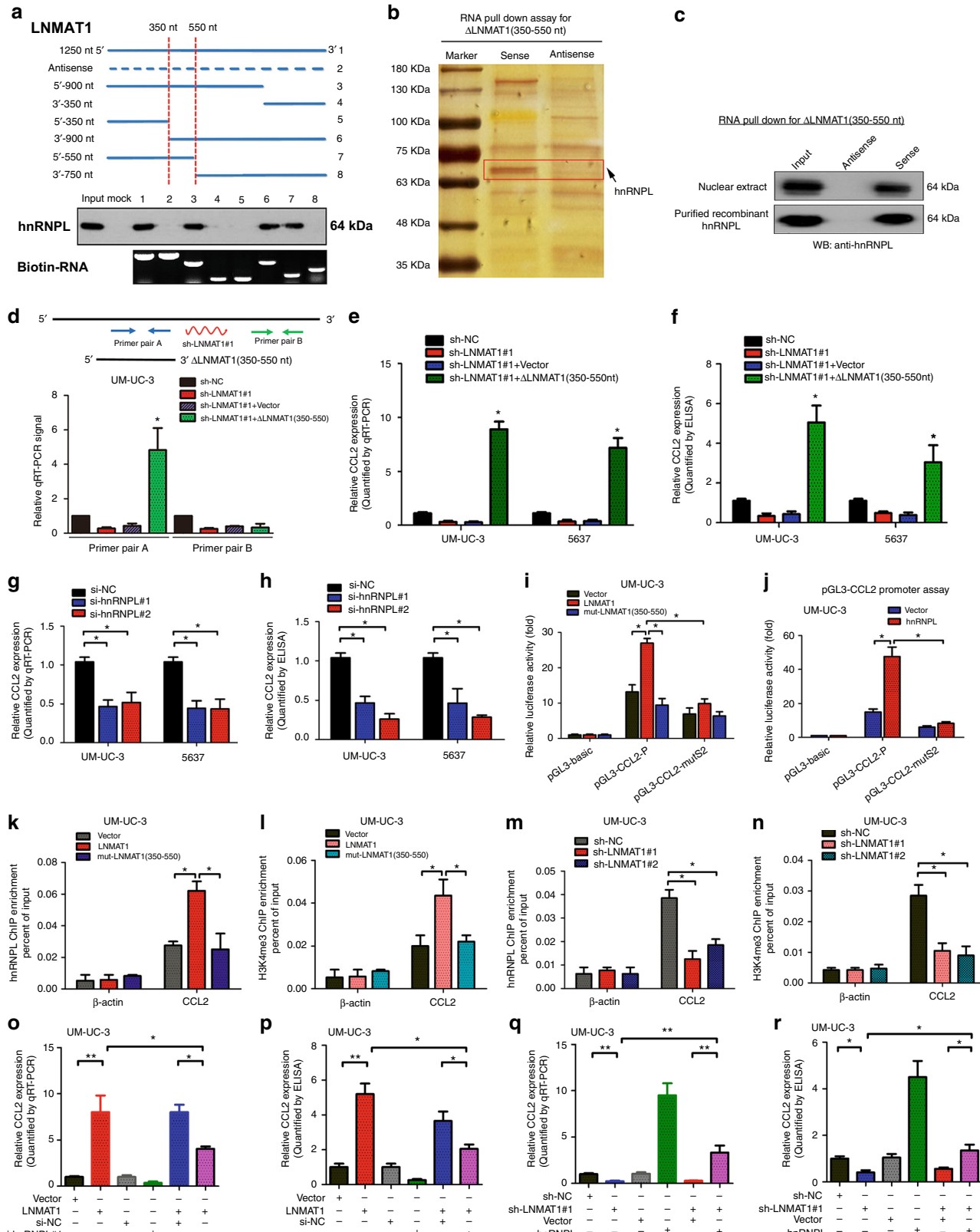

tumors formed by *LNMAT1*-transduced cells in the LNs, which led to prolonged survival times in the tumor-bearing nude mice, suggesting that the CCL2 blockade exerted promising anti-tumor effects in *LNMAT1*-overexpressing bladder cancer by inhibiting LN metastasis. Therefore, our findings uncover a molecular mechanism underlying CCL2 overexpression in the cancer cell

and suggest that a neutralizing antibody against CCL2 will serve as an effective therapeutic approach for patients with *LNMAT1*-overexpressing bladder cancers.

Our mechanistic studies also indicated that *LNMAT1* functions, at least in part, through its interaction with hnRNPL. hnRNPL contributes to gene expression through interactions with

**Fig. 6** *LNMAT1*/hnRNPL promotes H3K4me3 of the CCL2 promoter. **a** Serial deletions of *LNMAT1* were used in the RNA pull-down assays to identify the core regions of *LNMAT1* for the physical interaction with hnRNPL. **b** Silver staining image of proteins pulled down by the 5'-terminus of the truncated *LNMAT1* (350–550 nt). **c** The interaction between the truncated *LNMAT1* and hnRNPL was confirmed by RNA pull-down and western blotting with nuclear extract or purified recombinant hnRNPL. **d**–**f** Endogenous *LNMAT1* was depleted with a shRNA and the efficiency of the expression of the truncated *LNMAT1* was examined by qRT-PCR and ELISA analyses. Two-tailed *t*-tests was used. **g**, **h** qRT-PCR and ELISA analyses of CCL2 expression in the hnRNPL-silenced and control cell as indicated. Statistical significance was assessed using two-tailed *t*-tests and one-way analyses of variance (ANOVA) followed by Dunnett's tests for multiple comparison. **i**, **j** CCL2 wild-type (−2000 ~+121) or with *LNMAT1* binding site mutated promoter were subjected to luciferase reporter assays in *LNMAT1* or hnRNPL overexpressing cells. Two-tailed *t*-tests and one-way analyses of variance (ANOVA) followed by Dunnett's tests for multiple comparison. **k**–**n** ChIP-qPCR analysis of hnRNPL occupancy (**k**, **m**) and H3K4 methylation status (**l**, **n**) in the CCL2 promoter after overexpression/knock down of *LNMAT1* in UM-UC-3 cells as indicated. Two-tailed *t*-tests and one-way analyses of variance (ANOVA) followed by Dunnett's tests for multiple comparison. **o**–**r** qRT-PCR and ELISA analyses of CCL2 expression in the hnRNPL knockdown (**o**, **p**) or overexpression (**q**, **r**) or on *LNMAT1*-induced CCL2 expression in UM-UC-3 cells. Two-tailed *t*-tests and one-way analyses of variance (ANOVA) followed by Dunnett's tests for multiple comparison. The error bars represent standard deviations of three independent experiments. $*p < 0.05$ and $**p < 0.01$

the alternative splicing isoforms of different targets[49,50], and hnRNPL might also be involved in transcriptional regulation. Li et al. found that hnRNPL interacts with lncRNA-THRIL to regulate TNF-α transcription in THP-1 monocytes[51]. Fitzgerald et al. reported that hnRNPL directly binds to lncRNA-EPS and alters immune response genes (IRG) expression at the transcription level[27]. Here, we proposed a model in which *LNMAT1* recruited hnRNPL to specific genomic loci by forming triplexes with CCL2 promoter sequences. Therefore, identification of hnRNPL as a functional binding partner of *LNMAT1* adds further support to the roles of hnRNPs in transcriptional regulation, expanding the role of these RNA binding proteins beyond their well-known functions in mRNA processing.

Recently, selected RNA-motif small molecules were designed to target the folded structures of oncogenic noncoding RNAs and have shown significant antitumor effects in vivo by inhibiting cancer cells while leaving normal cells unaffected[52,53]. Our results show that *LNMAT1* silencing plays an important role in counteracting lymphatic metastasis, which indicates that *LNMAT1* might be a therapeutic target for cancer therapies. Lymphatic metastasis in bladder cancer patients indicates a poor prognosis[41,42]. The life expectancy of patients with bladder cancer is determined by metastatic dissemination that initially proceeds from lymphatic vessels to LNs via the lymphatic system and then eventually proceeds from blood vessels to distant organs[30]. Although the presence of lymphangiogenesis in the regional LN is a key prognostic survival marker for patients, diminishing lymphangiogenesis remains challenging. Thus, small-molecule-mediated ablation of the function of *LNMAT1* would provide a therapeutic strategy for the lymphatic metastasis of human cancer.

In this study, a popliteal LN metastasis model, in which bladder cancer cells were injected into the footpad of a mouse was used. Although footpad injection is the sensitive and quantitative method of measuring lymphatic metastasis in vivo, there are multiple limitations to this model. First, the microenvironment of the footpads is quite different from that of the bladder. Moreover, the intratumoral interstitial fluid pressure (IFP) is commonly much higher than that in the surrounding host tissues, which influences the different lymphatic fluid reflux and filtration rates, causing some differences in the lymphatic metastasis results[54]. In addition, the BALB/c nude mouse model lacked effective adaptive immunity, which is commonly used for studies to test pharmacological treatment of human tumor xenografts[55].

In summary, our study provides solid evidence supporting the hypothesis that *LNMAT1* overexpression is clinically and functionally relevant to the lymphatic metastasis of human bladder cancer via CCL2-mediated modulation of the tumor microenvironment. Understanding the precise role of *LNMAT1* in the LN metastasis of bladder cancer and in activation of the CCL2/

TAMs axis will not only increase our knowledge of lncRNA-induced LN metastasis but also enable the development of a therapeutic strategy for bladder cancer LN metastasis.

## Methods
**Patients and clinical samples**. A total of 266 formalin-fixed, paraffin-embedded bladder cancer specimens were collected from patients who underwent surgery at Sun Yat-sen Memorial Hospital of Sun Yat-sen University (Guangzhou, China) between February 2006 and March 2016. All samples were immediately snap-frozen in liquid nitrogen and stored at −80 °C until required. Two pathologists pathologically confirmed each sample by HE staining. Ethical consent was approved by the Committees for Ethical Review of Research involving Human Subjects at Sun Yat-sen University. Written informed consent was obtained from each patient before sample collection. The clinicopathological characteristics of the patients are summarized in Supplementary Table 2.

**Antibodies and reagents**. The following antibodies were used: anti-LYVE-1, Abcam (ab14917), for IHC; anti-F4/80, Abcam (ab6640), for IHC; anti-hnRNPL, Abcam (ab6106), for immunoblot, RIP and ChIP; anti-hnRNPQ, Abcam (ab184946), for immunoblot; anti-hnRNPA1, Abcam (ab4791), for immunoblot; anti-hnRNPL, Abcam (ab133607), for immunoblot; anti-CD68, Cell Signaling Technology, #76437, for IHC and flow cytometry; anti-CD206, Abcam (ab64693), for flow cytometry; anti-CCL2, Abcam (ab9669), for immunoblot and IHC; CCL2-neutralizing antibody (#554440, BD Biosciences, NJ, USA); anti-GAPDH, Cell Signaling Technology, #5174, for immunoblot; anti-H3K4me3, Abcam (ab8580), for immunoblot and ChIP; anti-VEGF-C, Abcam (ab9546), for immunoblot and IHC; and VEGF-C-neutralizing antibody (pV1006R-r, Angio-Proteomie, Boston, MA, USA). Control mouse IgG, control rabbit IgG, anti-RNA pol II, and anti-snRNP70 were provided in the EZ-Magna RIP kit or EZ-Magna ChIP A/G kit (Millipore). DAPI (Thermo Scientific, 62247) and Alexa Fluor™ 555 Phalloidin (Invitrogen™) were also used.

**Cell lines and cell culture**. The human bladder cancer cell lines UM-UC-3 and 5637 were obtained from the American Type Culture Collection (ATCC, Manassas, VA, USA). UM-UC-3 cells were cultured in Dulbecco's modified Eagle's medium (DMEM), and 5637 cells were cultured in RPMI 1640 (Gibco, Shanghai, China). All media contained 10% fetal bovine serum (FBS), 100 U/ml penicillin and 100 μg/ml streptomycin (HyClone, Thermo, USA). All cells were cultured in an incubator with 5% $CO_2$ at 37 °C.

**Cell isolation and cell differentiation**. The human monocytes were isolated from buffy coats prepared from healthy volunteer donors. Peripheral blood monocytes isolated by Ficoll-Hypaque (Pharmacia Corporation, Peapack, N.J.) for 50 min at 400 g density gradient centrifugation were seeded at $2 \times 10^6$/ml in 24-well plates in RPMI 1640 medium supplemented with 10% heat inactivated human AB serum, 50 U of penicillin/ml, 50 U of streptomycin/ml, 2 mM L-glutamine, and 100 ng/ml human M-CSF to stimulate macrophage differentiation. After 6 days of culture, non-adherent cells were removed by repeated gentle washing with warm medium. More than 95% of the adherent cells obtained with this technique were CD14+ macrophages. For in vitro activation, monocyte-derived macrophages at $2 \times 10^6$ cells/l were treated for 1 day with 25 μg/ml lipopolysaccharide (LPS, Sigma, Saint Louis, Missouri) in order to obtain M1-polarized macrophages or 45 ng/ml recombinant human interleukine-4 (IL-4, R&D, Minneapolis, MN) for M2-polarized macrophages. The differentiation of monocytes into macrophages was confirmed by flow cytometry. Cells were thoroughly washed in PBS before incubating for another 24 h in non-supplemented RPMI media to produce conditioned media to be used in subsequent in vitro assays.

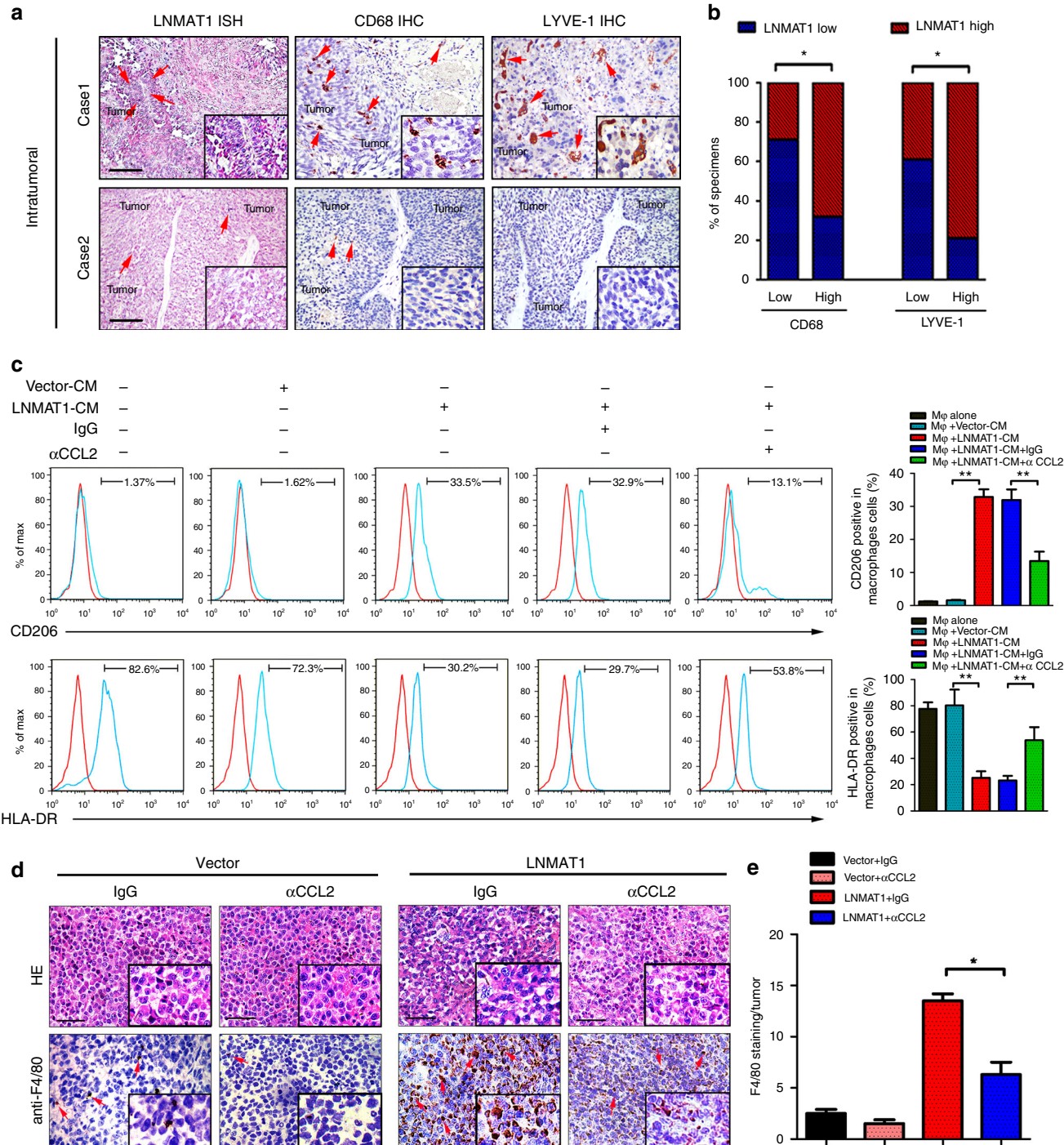

**Fig. 7** *LNMAT1*-upregulated CCL2 activates TAMs. **a**, **b** Representative images and percentages of bladder cancer tissues with high or low levels of CD68-positive cells and LYVE-1-positive cells in the intratumoral tissues with different *LNMAT1* expression levels. *LNMAT1* expression levels were quantified by ISH, macrophage density was quantified by IHC using anti-CD68 antibody and microlymphatic vessel density was quantified by IHC using the anti-LYVE-1 antibody. Two representative cases are shown. Statistical significance was assessed by $\chi^2$ test. Scale bars: 100 μm. **c** Flow cytometric analysis and quantification of the expressions of CD206/HLA-DR in macrophages treated with medium collected from the indicated cells for 24 h. CCL2-neutralizing treatment was carried out for an additional 18 h. Statistical significance was assessed by two-tailed *t*-tests. **d**, **e** Representative images (**d**) IHC staining evaluating macrophages with anti-F4/80 as indicating with red arrows and histogram analysis (**e**) in primary tumor from the footpads of nude mice. Scale bars: 100 μm. Statistical significance was assessed by two-tailed *t*-tests. The error bars represent standard deviations of three independent experiments. *$p < 0.05$, and **$p < 0.01$

**ISH and IHC analysis**. ISH and IHC were used to examine *LNMAT1* expression in paraffin-embedded primary carcinomas and xenograft tumor specimens from nude mice. After deparaffinization and rehydration, the samples were treated with a peroxidase-quenching solution. Then, 20 μg/ml proteinase K was added to digest the tissues, which were then fixed in 4% paraformaldehyde and hybridized at 42 °C overnight with a double-(5′ and 3′)-digoxin (DIG)-labeled locked nucleic acid (LNA)-modified *LNMAT1* probe. The samples were subsequently incubated overnight at 4 °C with an anti-digoxin monoclonal antibody conjugated to alkaline phosphatase (Roche). After staining with nitro blue tetrazolium/5-bromo-4-chloro-3-indolylphosphate (Roche), we observed and analyzed the sections. We used the

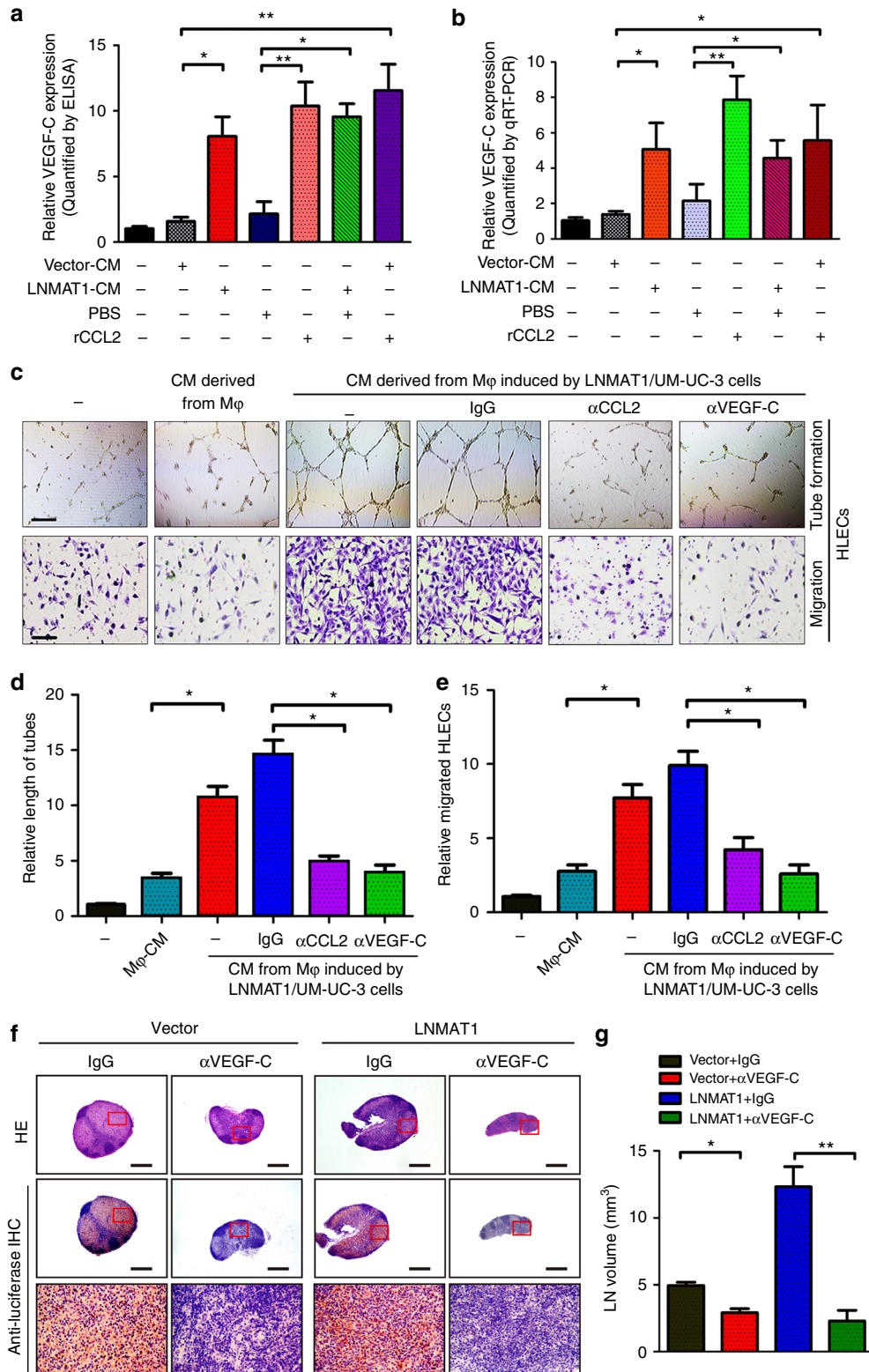

**Fig. 8** *LNMAT1*-upregulated CCL2 induces LN metastasis of bladder cancer. **a**, **b** ELISA (**a**) and qRT-PCR (**b**) analyses of VEGF-C expression in TAM-like macrophages induced by conditioned medium collected from the indicated cells. One-way ANOVA followed by Dunnett's tests for multiple comparison. **c**–**e** Representative images (**c**) and quantifications (**d**, **e**) of tube formation and Transwell migration by HLECs treated with conditioned medium collected from macrophages induced by *LNMAT1* from UM-UC-3 cells. HLECs were cultured with conditioned medium derived from TAM-like macrophages induced by the indicated cancer cells. Scale bars: 100 μm. Two-tailed *t*-tests and one-way analyses of variance (ANOVA) followed by Dunnett's tests for multiple comparison. **f** Representative images of HE and IHC staining confirming the LN status (*n* = 16). Scale bars: black, 500 μm; red, 100 μm. **g** Volume quantification of popliteal LN metastasis after inhibition of VEGF-C with neutralizing antibody. Two-tailed *t*-tests. The error bars represent standard deviations of three independent experiments. *$p < 0.05$ and **$p < 0.01$

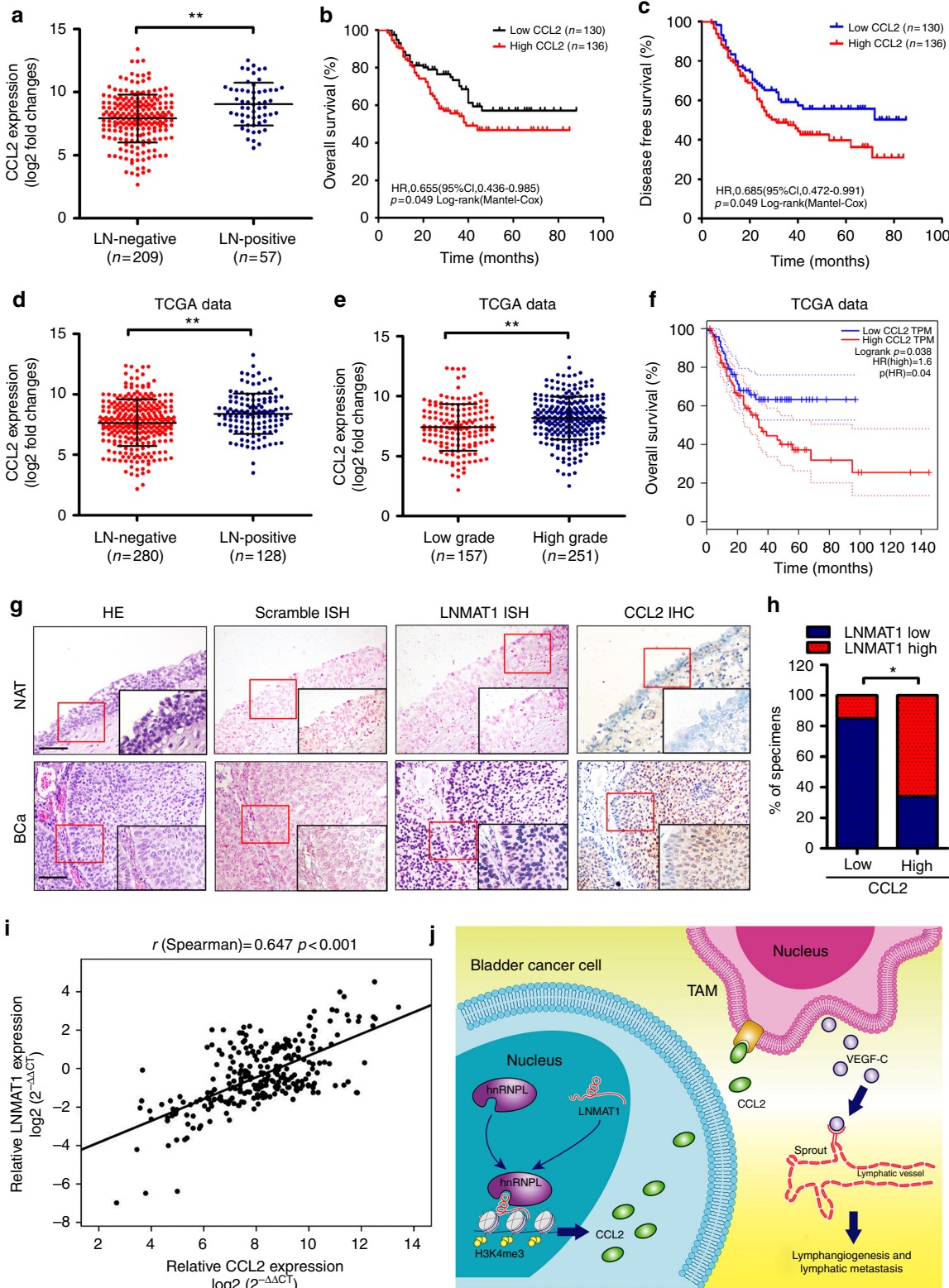

**Fig. 9** Clinical relevance of *LNMAT1*/CCL2/lymphangiogenesis in bladder cancer. **a** Correlation of CCL2 expression in bladder cancer tissues ($n = 266$) assessed by qRT-PCR with LN status. The nonparametric Mann–Whitney $U$-test was used. **b**, **c** Kaplan–Meier analysis for OS and DFS of bladder cancer patients with low vs. high expression of CCL2. The median CCL2 expression was used as the cutoff value. **d**–**f** TCGA data showed that *LNMAT1* overexpression correlated with LN metastasis (**d**), high grade (**e**), and poor prognosis (**f**). **g**–**h** The *LNMAT1* levels were positively correlated with CCL2 expression in the bladder cancer specimens ($n = 266$; $p < 0.001$). Scale bars: 100 μm. Statistical significance was assessed by $\chi^2$ test. **i** Correlation analysis of *LNMAT1* and CCL2 mRNA expression by qRT-PCR. **j** Illustrative model showing the proposed mechanism by which *LNMAT1* promotes lymphatic metastasis in bladder cancer via CCL2-dependent TAMs recruitment. The error bars represent standard deviations of three independent experiments. *$p < 0.05$ and **$p < 0.01$

double-(5′ and 3′)-DIG-labeled scrambled probe as the NC. The sequences of the probes used for ISH are shown in Supplementary Table 7.

The percentage of tumor cells with positive staining was designated follows: 0 (no positive), 1 (0–10% positive), 2 (10–30% positive), 3 (30–70%), and 4 (over 70%). The staining intensity was graded as follows: 1 (no staining), 2 (weak staining, light yellow for IHC, light blue for ISH), 3 (moderate staining, brown for IHC, moderate blue for ISH) and 4 (strong staining, brown red for IHC, strong blue for ISH). The staining index (SI) was calculated with possible scores of 0, 1, 2, 3, 4, 6, 8, 9, 12, and 16. Then, the median value, which is SI = 8, was chosen as the cut off value. Therefore, samples with an SI ≥ 8 had high expression and samples with an SI < 8 had low expression. To avoid evaluation biases, ISH and IHC analyses were independently performed by two experienced pathologists who were blinded to the tissue information. Cases with discrepancies were jointly re-evaluated until a consensus was reached. Images were visualized using a Nikon Eclipse Ti (Japan) microscope system and processed with Nikon software.

**RNA pull down and RNA immunoprecipitation (RIP)**. Full-length *LNMAT1* and antisense sequences were prepared by in vitro transcription using a Transcript Aid T7 High Yield Transcription Kit (Thermo Scientific). The sequences were treated with RNase-free DNase I and purified with the GeneJET RNA purification kit (Thermo Scientific). RNA pull down assays were performed with the Magnetic RNA-Protein Pull down Kit (Thermo Scientific) according to the manufacturer's instructions. Nuclear extracts were prepared with the NE-PER Nuclear Protein Extraction Kit (Thermo Scientific). Three micrograms of biotin-labeled RNA and 1 mg of nuclear extract were used in each pull down assay. The retrieved protein was separated on polyacrylamide gel electrophoresis (PAGE) gels and visualized with a silver staining kit (Thermo) or by standard immunoblotting. The bands specific to *LNMAT1* were excised, and proteomics screening was accomplished by MS analysis on a MALDI–TOF instrument (Bruker Daltonics). The uncropped gels are summarized in Supplementary Figure 16.

The RIP assay was performed using the EZ-Magna RIP kit (Millipore, MA, USA). Briefly, $1 \times 10^7$ cells were harvested and lysed with RIP lysis buffer with one freeze–thaw cycle. Cell extracts were co-immunoprecipitated using anti-hnRNPL (Abcam, MA, USA), and the retrieved RNA was subjected to qRT-PCR analysis. Normal mouse IgG was used as the NC. For qRT-PCR analysis, U1 RNA was used as a non-specific control.

**ChIP and ChIRP analysis**. The ChIP experiments were performed using the EZ-Magna ChIP A/G kit (Millipore, MA, USA). A total of $1 \times 10^6$ cells were fixed in 1% formaldehyde at room temperature for 10 min, and the nuclei were isolated with nuclear lysis buffer supplemented with a protease inhibitor. The chromatin DNA was sonicated and sheared to lengths between 100 and 200 bp. The sheared chromatin was immunoprecipitated at 4 °C overnight using an anti-hnRNPL antibody (Abcam, MA, USA) or anti-H3K4me3 antibody (Abcam, MA, USA). Normal mouse IgG was used as the NC, and an anti-RNA pol II antibody (Millipore) was used as the positive control. The ChIP-qPCR primers are listed in Supplementary Table 6.

The Magna ChIRP RNA Interactome Kit was purchased from Millipore (Millipore, MA, USA) and used according to the manufacturer's instructions. Briefly, the probes were designed using a single-molecule FISH online designer, were biotin-labeled at the 3′end and were divided into an "odds" or an "even" groups. A total of $2 \times 10^7$ bladder cancer cells were cross-linked for each hybridization reaction. Then, the cell lysate was sonicated to shear the chromatin into 100–200 bp fragments. The sonicated cell lysates were hybridized with a mixture of biotinylated DNA probes for 4 h at 37 °C. Then, the binding complexes were recovered using streptavidin-conjugated magnetic beads. Finally, DNA, RNA and protein were eluted and purified from the beads. The probes used in the ChIRP assay are listed in Supplementary Table 7.

**FRET assay**. The pyrimidine strand contained an FAM label on its 5′ end. The corresponding 13-mer TFO1 was labeled on its 5′ end with 5-carboxytetramethylrhodamine (TAMRA). FAM and TAMRA form a FRET pair with an R0 of 49–54 Å. Triplex formation was assessed in binding buffer (20 mM HEPES pH 7.5, 50 mM Na-acetate, 10 mM $MgCl_2$). For our experiment, 500 nM double-stranded DNA and 2500 nM TFO were mixed, annealed by heating to 55 °C for 10 min and incubated at 37 °C for 10 h. The fluorescence intensities were measured with a Molecular Device M5 Plate Reader using an excitation of 470 nm and emission wavelengths of 480 and 690 nm. Each FRET experiment was repeated three times using different batches of transcripts.

**CD spectroscopy**. CD spectroscopy was performed on a Chirascan. Briefly, CD spectra were recorded for the *LNMAT1* ssRNA TFO (2.2 μM) and the three different dsDNA TTS oligos (corresponding to the CCL2 promoter sequences predicted to be associated with *LNMAT1*, 2.2 μM each) separately and together with a 1:1 mix of the two (2.2 μM TFO and 2.2 μM TTS) in 1× triplex-forming buffer (10 mM Tris pH 7.5, 25 mM NaCl and 10 mM $MgCl_2$). LongTarget was used to predict the TFOs and TTS. For comparisons, samples with a control sRNA were included. The mixed samples or individual RNAs and dsDNAs were equilibrated for approximately 1 h at 30 °C. The measurements were performed at room

temperature in a 1-mm cuvette using 200 μl of solution and kept at room temperature before measurement. The spectra of the buffer solution were subtracted to correct for baseline artifacts. CD spectra were normalized in units of molar ellipticity.

**Popliteal lymphatic metastasis model**. The BALB/c nude mice (4–5 weeks old, 18–20 g) were purchased from the Experimental Animal Center, Sun Yat-sen University (Guangzhou, China). All experimental procedures were approved by the Institutional Animal Care and Use Committee of Sun Yat-sen University. The footpads of 16 mice were inoculated with 100 μl PBS suspensions of bladder cancer cells that were transduced with *LNMAT1*-luc, vector-luc, sh*LNMAT1*-luc or sh-NC-luc. One week after the tumor cell inoculations, the nude mice were randomly selected for treatment by intraperitoneal injection with the PBS control or the CCL2 neutralizing antibody three times per week. Lymphatic metastasis was monitored and imaged with a bioluminescence imaging system (PerkinElmer, IVIS Spectrum Imaging System) 4 weeks after the injections. The tumor growth and that lymphatic metastasis were checked when the control tumors reached the same size as the experimental tumors. The primary tumors and popliteal LNs were enucleated and paraffin embedded. Serial 4.0-mm sections were obtained and analyzed by IHC. Images were captured using a Nikon Eclipse 80i system with NIS-Elements software (Nikon, Japan). The uncropped pictures are summarized in Supplementary Figure 15.

**Subcutaneous tumorigenicity and tail vein injection assay**. For the subcutaneous tumorigenicity assay, $1 \times 10^6$ cells transduced with vector-luciferase or *LNMAT1*-luciferase in 200 μl PBS were inoculated subcutaneously. For the tail vein injection assay, UM-UC-3 cells were injected into tail vein of each mouse. Lung metastases were monitored and quantified with a bioluminescence imaging system 4 weeks post-injection. The tumors and lungs were excised and fixed in formalin overnight and embedded in paraffin, from which sections were stained with HE by pathologists. The qRT-PCR and ISH were used to confirm *LNMAT1* expression. For bioluminescence imaging, mice received luciferin (300 mg/kg, 10 min before imaging) and were anesthetized with 3% isoflurane, imaged in an IVIS spectrum imaging system.

**HLECs tube formation assay and transwell assays**. Serum-free media obtained from the co-culture of tumor cells and macrophages were concentrated ten-fold using ultrafiltration spin columns (Millipore, Billerica, MA, USA). HLECs were then seeded into six-well plates (pre-coated with Matrigel) containing concentrated media and incubated for 12 h. The resulting lymphatic tubes were photographed using an inverted microscope and quantified by measuring the number and area of the completed tubule structures.

Cell invasion was evaluated using Transwell chambers (BD Biosciences, MA, USA) according to the manufacturer's protocol. Cells were subjected to 24 h of serum deprivation and seeded to 24-well Matrigel-coated Transwell plates in serum-free medium. The bottom wells were filled with complete medium. After incubation for 24 h with 5% $CO_2$ at 37 °C, the cells remaining in the upper chambers were scraped off, and the invading cells were fixed with 4% paraformaldehyde and stained with crystal violet. Five random fields were microscopically examined, and the number of cells was determined.

**Immunoblotting analysis**. The cells were lysed using the protein extraction reagent RIPA buffer (Pierce, Rockford, IL, USA) supplemented with a protease inhibitor cocktail (Roche, Pleasanton, CA, USA). The proteins were extracted, separated by SDS-PAGE on 10% gels and transferred to polyvinylidene fluoride membranes. Then, the membranes were blocked with 5% bovine serum albumin (BSA) and incubated in solutions containing primary antibodies overnight at 48 °C. The membranes were incubated with corresponding secondary antibodies and visualized using an ECL chemiluminescence kit. The uncropped blots are summarized in Supplementary Figure 16.

**Statistics**. All quantitative data are presented as the mean ± standard deviation from at least three independent experiments. The chi-square test ($\chi^2$ test) for non-parametric variables and Student's *t*-test or one-way analysis of variance (ANOVA) for parametric variables (two-tailed tests) were used to identify statistically significant data. OS and DFS were evaluated using the Kaplan–Meier method. All statistical analyses were conducted using SPSS v.13.0 (SPSS Inc., Chicago, IL, USA), and *p*-values < 0.05 were considered statistically significant.

## Data availability

The next-generation sequencing data used in the study (GSE106534) are available in a public repository from NCBI (https://www.ncbi.nlm.nih.gov/geo/query/acc.cgi?acc=GSE106534). The next-generation sequencing data used in the study (GSE106637) are available in a public repository from NCBI (https://www.ncbi.nlm.nih.gov/geo/query/acc.cgi?acc=GSE106637). All relevant data are available from the authors.

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

## Acknowledgements

The authors thank Prof. J.X. Zhang, Department of Medical Statistics and Epidemiology, School of Public Health, Sun Yat-Sen University, Guangzhou, China, for statistical advice and research comments. This study was funded by the National Natural Science Foundation of China (Grant No. 81802530, 81572514,81472384, 81472381, 81402106, 81772719, 81772728, 91740119, 91529301); Guangdong Medical Research Fund (A2018330); Science and Technology Program of Guangzhou (Grant No. 201604020156, 201604020177, 201707010116, 201803010098); National Natural Science Foundation of Guangdong (Grant No. 2018A030313564, 2018B030311009, 2016A030313321, 2015A030311011, 2015A030310122, S2013020012671, 07117336,10151008901000024). Yixian Youth project of Sun Yat-sen Memorial Hospital (YXQH201812).

## Author Contributions

C.C., H.J. and L.T. participated in the study design. H.W., S.F., Z.Y., D.W. and W.B. performed the in vitro and in vivo experiments and data analyses. Z.G., L.H. and B.J. performed the clinical data analyses. Z.B. and J.N. performed the ISH and IHC experiments. C.Q., C.C. wrote the manuscript. All the authors read and approved the final manuscript.

## Additional information

**Competing interests:** The authors declare no competing interests.

