## [Peer Review File · Nature Communications]

Reviewers' comments:

Reviewer #1 (Expertise: BC mouse models, Remarks to the Author):

This is an interesting study that addresses the role of long non coding RNAs in bladder cancer metastasis. The importance of the study is based on the relatively little that is known about the mechanisms underlying metastasis and in particular the role of RNAs. The current study identifies a long non-coding RNA that is associated with metastasis. The study is very interesting but some of the data are not clearly presented or entirely convincing.

First of all -- why choose this long non coding RNA from among those identified? This should be clarified.

Should explain why the popliteal metastasis assay is relevant for these studies.

The connection to lymphogenesis is very difficult to evaluate.

Need to demonstrate that the LNMAT1 is specific expressed. There is limited data confirming the expression.

Figure 1: (1) Specify the datasets being used. (2) Panel J — the staining is not very evident. (3) Panel C is too small to read.

Figure 2: (1) What is the advantage of using the popliteal metastasis assay. (2) Panel D is illegible. Whatever data is intended to be conveyed is not evident. (3) Panel F should have the actual p-values.

Figure 3: (1) Legend does not match the images. (2) the metastases are not confirmed beyond the H&E images.

Figure 4: (1) P-values should be included on the Kaplan Meier images. (2) Panel I - images are too small to see the data - suggest showing an inset for the data.

Figure 5: Data are too difficult to evaluate - not clearly legible. Data for supporting a triplex are not sufficiently convincing.

Reviewer #2 (Expertise: CCL2, TAM, metastasis, Remarks to the Author):

In the manuscript "Long noncoding RNA LNMAT1 promotes lymphatic metastasis of bladder cancer via CCL2-dependent macrophage recruitment" the authors found and characterized a novel long noncoding RNA (LMAT1) implicated in lymphatic dissemination of bladder cancer cells. Additionally, the authors propose a mechanism where LNMAT1 upregulates CCL2 in cancer cells leading to a higher recruitment of macrophages, which promotes lymphangiogenesis and metastasis.

The results are interesting and novel, the mechanism of CCL2 activation by LNMAT1 is well-characterized and the correlation with human data is remarkable. However, the manuscript seems incomplete, with a number of significant issues that need to be addressed, as described below.

Major comments:

1. Materials and methods information about certain experiments (e.g. wound healing) is missing. Moreover, "Methods" and "Supplementary Materials and Methods" seem to be exactly the same, whereas those 2 parts should complement each other.

2. The reviewer is not convinced about the cell isolation and differentiation protocols of human monocytes in macrophages. How is it possible to differentiate monocytes into macrophages without adding any differentiation factors (usually GM-CSF or M-CSF)? Furthermore, 24 hours of treatment with LPS or IL-4 are usually considered sufficient for macrophage polarization, so why did the authors decide to polarize them for 7 days? The authors also should mention the paper in which the protocol that they followed is reported. What kind of macrophages were used in experiments shown in Fig. 7 and Fig. 8? Why do the authors claim they are TAMs if they were not

isolated from tumors?

3. Could the authors assess the influence of corresponding treatments (Fig. 7 C) on the expression of at least one more M2 marker and two chosen M1 markers (FACS)? For the assessment of capillary formation, could the authors use CM from TAMs sorted from murine tumors (overexpressing and silenced for LNMAT1)?

4. Except for microlymphatic vessel density, could the authors also assess the presence of big LVs in murine tumors sh-NC vs sh-LNMAT1 (Fig. 2H,I), as usually such bigger vessels participate in lymphatic metastasis? Moreover, the authors quantified signal of "LYVE-1 positive cells", and attributed it to lymphatic vessels. How do the authors know that they are not macrophages? Could the authors use double fluorescence staining (F4/80 and LYVE-1 or F4/80 and VEGFR3) in order to quantify LVs exclusively?

5. The authors demonstrate the presence of increased metastasis upon LNMAT1 overexpression in both tail vein and tumor inoculation experiments. Thus, does it observation rely more on increased tumoral lymphangiogenesis or increased extravasation of cancer cells? Could the authors assess the metastasis levels from primary tumors upon the blockage of lymphangiogenesis (e.g. with anti-VEGF-C)? Moreover, could the authors assess the migration of cancer cells (with and without LNMAT1 overexpression) on the invasion through lymphatic endothelium monolayer in vitro?

Minor comments:

1. The manuscript has a many grammar mistakes that need to be corrected (e.g. "lncRNAs regulate chemokine activation by interact with chromatin" should be "lncRNAs regulate chemokine activation by interaction with chromatin"). Please revisit the whole manuscript.

1. Could the authors indicate what was the LN status in high-grade muscle invasive bladder cancers (MIBC) used for NGS analysis (Fig. 1A).

2. The authors demonstrate the effects of LNMAT1 on LN metastasis (Figure 4I). What are the effects of the corresponding treatments on distant organ metastasis, as this is a more clinically relevant question?

3. Could the authors assess the effect of CCL2 blockage in vitro (a migration or a wound healing assay with cancer cells)?

4. In Fig. 6G,H the cells lines were silenced for hnRNPLs, however in the text referring to this figure the authors state that the cells were transduced to express hnRNPLs. Which version is correct?

5. What are the p values in Fig. 2G and Fig. 4J?

6. As mentioned in Figure 1, three lncRNAs, (LNMAT1, CTD-2231H16.1 and BCAR4) were consistently upregulated both in the MIBC and LN-positive bladder cancer tissues (Figure 1C). Beside the significant overexpression of LNMAT1 in figure 1E, could the authors also display the expression levels of the other 2 lncRNAs in the supplemental figures?

7. The authors need to correct figure legend accuracy (e.g. no figure legend in Fig. 3G; Fig. 3D appears twice in the legend; what do the arrows in Fig. 4I indicate?).

8. Some figures do not appear in the order introduced in the text.

Reviewer #3 (Expertise: lncRNA, epigenetics, Remarks to the Author):

The manuscript by Chen et al. focused on the role of a long non-coding RNA LNMAT1 in lymphnode

metastasis from bladder cancer. The authors showed evidence that LMNAT1 is associated with advanced grade of bladder cancer, poor prognosis and lymphatic infiltration. The authors further used a somewhat artificial tumor model, in which human bladder cancer cells were injected into the footpads of nude mice, to examine the function on lymphatic infiltration. They concluded that LMNAT1 enhanced lymphatic infiltration through a CCL2 dependent mechanism. Mechanistically, the authors demonstrate that LMNAT1 enhances CCL2 expression from cancer cells, which is regulated possibly through direct binding of LMNAT1 to CCL2 promoter, recruiting hnRNPL, leading to H3K4me3 mark deposition on CCL2 promoter. In the authors' model, the increase of CCL2 in cancer cells leads to increased macrophage recruitment, and possibly M2-like polarization, resulting in increased VEGF-c expression to enhanced lymphangiogenesis.

This study is an overall interesting story. The major novel contribution is the role of LMNAT1 in lymphatic infiltration by bladder cancer cells, and the mechanism through which LMNAT1 regulates CCL2. There are a lot of data ranging from human clinical specimens to mouse models to molecular mechanisms. The mechanistic studies, particularly the regulation of CCL2 expression, were well done, and convincing to me. The correlation between CCL2 and LMNAT1 in human cancer data is also supporting the mechanism and very nice. However, a few major concerns on the in vivo experiments exist that need to be properly addressed.

1. In the in vivo model, the authors showed that modulating LMNAT1 leads to changes in lymph node size and lymphatic infiltration. However, they did not show any data on the primary tumor size. An equally possible alternative explanation of their data is that LMNAT1 regulates primary tumor size, and larger primary tumor leads to increased lymphatic infiltration. Judging from the luciferase images (e.g. fig 4i), the primary tumor size seems to correlate with LMNAT1 modulation.

2. The critical experiment in which anti-CCL2 antibody treatment reduces lymphatic infiltration lacks key controls. This experiment is key to pinpoint a functional role of tumor-cell-derived CCL2 in the lymphatic infiltration process. Currently, anti-CCL2 is only used in mice with cells overexpressing LMNAT1. But it is equally possible that anti-CCL2 is inhibiting CCL2 from other non-cancer cell types in vivo—if true, one would predict that anti-CCL2 would be effective in control bladder cancer cells as well.

3. Follow up with the point 2 above, another experiment that can help pinpoint a role of tumor-cell derived CCL2 is to perform CCL2 knockdown in cancer cells, rather than using antibody—antibody cannot discriminate the source of CCL2.

4. The in vivo tumor model has limitations—it is human bladder cells injected in a non-physiological site (foot pad), with the lack of effective adaptive immunity (using nude mice). I think the authors should at least discuss these limitations.

Minor:

1. The authors claim the use of "in vitro synthesized hnRNPL", but I couldn't find this info in the methods. Is this truly "synthesized", or it is purified recombinant protein expressed in bacteria or another host?

2. The histology pictures (e.g. Fig 2h, 2i, 7a, 7b) are too small with low resolution of details. Not very easy to see even after expanding on computer screen. Maybe including the large versions in supplement?

Reviewer #1:

This is an interesting study that addresses the role of long non-coding RNAs in bladder cancer metastasis. The importance of the study is based on the relatively little that is known about the mechanisms underlying metastasis and in particular the role of RNAs. The current study identifies a long non-coding RNA that is associated with metastasis. The study is very interesting but some of the data are not clearly presented or entirely convincing.

1: First of all -- why choose this long non-coding RNA from among those identified? This should be clarified.

Response: We thank the reviewer for raising this important point and the reviewer's comment is greatly appreciated. In this study, next-generation sequencing (NGS) was performed in 5 paired high-grade muscle invasive bladder cancer (MIBC) tissues and normal adjacent tissues (NATs), as well as in another 5 LN-positive and LN-negative bladder cancer tissues. Statistical analysis revealed that 32 lncRNAs were significantly upregulated and 35 lncRNAs were downregulated in the high-grade MIBC tissues compared with the NATs, and 35 lncRNAs were upregulated, but 25 lncRNAs were downregulated, in the LN-positive bladder cancer tissues compared with the LN-negative samples by more than 5-fold, respectively (Fig. 1a-b). Among these identified lncRNAs, 3 lncRNAs, including *LNMAT1*, CTD-2231H16.1 and BCAR4, were consistently upregulated in both the MIBC and LN-positive bladder cancer tissues (Fig. 1c). Moreover, only *LNMAT1* (RefSeq accession number: NR_122112.1) was most significantly overexpressed in the larger cohort of 266-case of bladder cancer tissues and paired normal adjacent tissues (NATs) (Fig. 1d and Supplementary Fig. 1a-c). Additionally, statistical analysis showed a strongly correlation between *LNMAT1* expression with LN metastasis status, which was further confirmed by analyses of The Cancer Genome Atlas (TCGA) database (Fig. 1e-f and Supplementary Fig. 2g-h). All these results suggest that *LNMAT1* may play a crucial role in LN metastasis of bladder cancer, which prompted us to choose *LNMAT1* for further investigation. The abovementioned descriptions have been incorporated into the revised manuscript.

2: Should explain why the popliteal metastasis assay is relevant for these studies. The connection to lymphogenesis is very difficult to evaluate.

Response: The reviewer's point is well taken. As suggested by the reviewer, the relevance of popliteal metastasis model has been discussed in the revised manuscript as the following: In

the clinic, the most common sites of lymph node metastases of bladder cancer are the pelvic lymph nodes, including the obturator nodes, the external iliac nodes and the common iliac lymph nodes. The lymphatic metastasis of bladder cancer is directional and the common iliac lymph nodes are considered as the next station to the obturator and external iliac nodes. In the current study, popliteal lymph node metastasis model was established in BALB/c-nude mice by injecting the footpads with bladder cancer cells. Lymph drainage in the footpad is unidirectional with the popliteal, which leads to lymph draining directly up to the popliteal lymph node and subsequently to the external iliac nodes and the common iliac lymph nodes, which are the common sites of lymph node metastases of bladder cancer. Therefore, the popliteal lymph node metastasis model may simulate the functional characteristics of lymphatic metastasis of bladder cancer typically seen in the clinic. Furthermore, the well-defined lymph drainage from footpad injections has enabled more sensitive and quantitative *in vitro* measurements of lymphatic metastasis. Thus, the popliteal lymph node metastasis model of injecting cells into mouse footpads has been widely used in many previously published studies and has proved reliable and suitable in many cases^{1,2,3}. The abovementioned descriptions have been added in the revised manuscript (as shown in page 6 in the text, lines 16-22 and page 19 in the text, lines 1-9).

References

- 1) Ito T, *et al.* Pituitary tumor-transforming 1 increases cell motility and promotes lymph node metastasis in esophageal squamous cell carcinoma. *Cancer research***68**, 3214-3224 (2008).
- 2) He W, *et al.* Long noncoding RNA BLACAT2 promotes bladder cancer-associated lymphangiogenesis and lymphatic metastasis. *J Clin Invest***128**, 861-875 (2018).
- 3) Liu L, *et al.* TBL1XR1 promotes lymphangiogenesis and lymphatic metastasis in esophageal squamous cell carcinoma. *Gut***64**, 26-36 (2015).

3: Need to demonstrate that the *LNMAT1* is specifically expressed. There is limited data confirming the expression.

Response: We thank the reviewer for this suggestion and the reviewer's point is well taken. The qRT-PCR and in situ hybridization (ISH) analysis showed that *LNMAT1* expression was marginally detected in normal bladder tissues and slightly increased in non-LN-metastatic bladder cancer tissues but strongly upregulated in LN-metastatic bladder cancer, suggesting that *LNMAT1* was more specifically overexpressed in LN-metastatic bladder cancer tissues (Fig. 1j). Moreover, analysis of tumor profiles in TCGA database showed that *LNMAT1* is

significantly overexpressed in various types of human cancers, such as bladder, thyroid, prostate, kidney, lung and liver cancers (Supplementary Fig. 2a-f). Consistently, analysis of tumor profiles in TCGA databases indicated that *LNMAT1* is overexpressed in LN-metastatic human cancers, such as thyroid carcinoma and kidney cancer (Supplementary Fig. 2g-h). Importantly, *LNMAT1* overexpression correlated with poor prognosis in many human cancers, such as thyroid carcinoma, kidney cancer, colon carcinoma and liver cancer (Supplementary Fig. 2i-p). These results suggest that *LNMAT1* is specifically expressed in LN-metastatic human cancer tissues and may play an oncogenic role in the progression and development of various human cancer types.

4: Figure 1: (1) Specify the datasets being used. (2) Panel J — the staining is not very evident. (3) Panel C is too small to read.

Response: We thank the reviewer for this suggestion and the reviewer's point are well taken:

(1) We are sorry that we did not write this point clearly in our originally submitted manuscript and appropriate modifications have been made in the revised manuscript. All the bladder cancer tissues used in Fig. 1 were our collected bladder cancer samples. The results analyzed in TCGA datasets was shown in Supplementary Fig. 2. Our collected bladder cancer samples shown in Fig. 1 include Fig. 1a-b: 5 paired high-grade muscle invasive bladder cancer (MIBC) tissues and normal adjacent tissues (NATs) and another 5 LN-positive and LN-negative bladder cancer tissues used for NGS analysis; Fig. 1d: 266 paired bladder cancer tissues and normal adjacent tissues. Fig. 1e: 195 high pathological grade samples and 71 low pathological grade samples; Fig. 1f: 57 samples with LN-metastasis and 209 samples with non-LN-metastasis. Fig. 1g: 57 bladder cancer with LN-metastasis samples compared with the paired primary tumors. The abovementioned results have been incorporated into the revised manuscript.

(2) As suggested by the reviewer, the images in the Panel J of Fig. 1 have been replaced by the higher resolution image in the revised manuscript.

(3) As suggested by the reviewer, the images in the Panel C of Fig. 1 have been replaced by a larger image in the revised manuscript.

5: Figure 2: (1) What is the advantage of using the popliteal metastasis assay. (2) Panel D is illegible. Whatever data is intended to be conveyed is not evident. (3) Panel F should have

the actual p-values.

Response: (1) We appreciate the reviewer's important point. In the clinic, the most common sites of lymph node metastases of bladder cancer are the pelvic lymph nodes, including the obturator nodes, the external iliac nodes and the common iliac lymph nodes. The lymphatic metastasis of bladder cancer is directional, and the common iliac lymph nodes are considered as the next station of the obturator and external iliac nodes. In our study, since lymph drainage in footpad is unidirectional with popliteal, which leads to lymph draining directly up to the popliteal lymph node and subsequently to the external iliac nodes and the common iliac lymph nodes, which are the common sites of lymph node metastases of bladder cancer. Thus, popliteal lymph node metastasis model may simulate the functional characteristics of lymphatic metastasis of bladder cancer typically seen in the clinic. Moreover, the well-defined lymph drainage from footpad injections has enabled more sensitive and quantitative *in vitro* measurements of lymphatic metastasis. Therefore, the popliteal lymph node metastasis model of injecting cells into mice footpads have been employed widely by numerous previously published studies, which has proved reliable and suitable^{1,2,3}. The abovementioned descriptions have been added to the revised manuscript.

(2) The reviewer's point is well taken. As required by the reviewer, the images in the Panel D of Fig.2 have been replaced by a higher resolution image in the revised manuscript.

(3) The reviewer's point is well taken. As required by the reviewer, the *P*-values have been added to the Panel F (please refer to Panel G) in Fig.2 and to the figure legends in the revised manuscript.

References

- 1) Ito T, *et al.* Pituitary tumor-transforming 1 increases cell motility and promotes lymph node metastasis in esophageal squamous cell carcinoma. *Cancer research***68**, 3214-3224 (2008).
- 2) He W, *et al.* Long noncoding RNA BLACAT2 promotes bladder cancer-associated lymphangiogenesis and lymphatic metastasis. *J Clin Invest***128**, 861-875 (2018).
- 3) Liu L, *et al.* TBL1XR1 promotes lymphangiogenesis and lymphatic metastasis in esophageal squamous cell carcinoma. *Gut***64**, 26-36 (2015).

6: Figure 3: (1) Legend does not match the images. (2) the metastases are not confirmed beyond the H&E images.

Response: (1) We apologize for the mistakes and thank the reviewer for pointing out the error. We have revised the figure legends and appropriate corrections have been made in the

revised manuscript.

(2) We thank for the reviewer's comment. As requested by the reviewer, we further confirmed the metastatic bladder cancer cells in the lung by IHC using an anti-luciferase antibody (Fig. 3f-g). Consistently, the luciferase signals in the lungs of mice injected with *LNMAT1* cells were much higher than those in mice injected with control cells, suggesting that *LNMAT1* overexpression strongly increased lung colonization by bladder cancer cells. The abovementioned results have been incorporated into the revised manuscript.

7: Figure 4: (1) P-values should be included on the Kaplan Meier images. (2) Panel I - images are too small to see the data - suggest showing an inset for the data.

Response: (1) We thank for the reviewer's comment. As requested by the reviewer, the P-values have been included on the Kaplan Meier images in Fig. 4k in the revised manuscript.

(2) The reviewer's point is well taken. As requested by the reviewer, the Panel I has been enlarged and appropriate inset data have been incorporated into Panel I of Fig. 4 in the revised manuscript.

8: Figure 5: Data are too difficult to evaluate - not clearly legible. Data for supporting a triplex are not sufficiently convincing.

Response: (1) We thank for the reviewer's comments and we apologize for the unclear images in our originally submitted manuscript. Appropriate modifications have been made and the higher resolution images have been added to the revised manuscript (Fig. 5).

(2) We thank for the reviewer's comment. To examine whether *LNMAT1* formed a DNA-RNA triplex with the CCL2 promoter, luciferase activity and ChIRP analysis were further performed. The promoter luciferase assay showed an obvious increase in the transcriptional activity of the construct contained region -200 to +121 bp than in the +1 to +121 bp region (Fig. 5c and Supplementary Fig. 8c). Moreover, a chromatin isolation by RNA purification (ChIRP) assay, which determines the exact locations of lncRNA binding sites on the chromatin¹, revealed that *LNMAT1* bound to -10 to -118 bp (referred to as CCL2-S2; Fig. 5d-f). Furthermore, analysis with LongTarget, a web-based tool designed to predict lncRNA-DNA binding motifs and binding sites via Hoogsteen and reverse Hoogsteen interactions², identified six potential triplex-forming oligonucleotides (TFOs) within

LNMAT1 and the corresponding triplex target sites (TTS) in the CCL2 promoter for possible pairing (Supplementary Table 5). Next, we synthesized the predicted TFO labeled with 5-carboxy tetramethyl-rhodamine (TAMRA) and TTS labeled with fluorescein amidite (FAM) and performed fluorescence resonance energy transfer (FRET) analysis. FRET showed an obvious increase in the TAMRA fluorescence intensity at 570-580 nm and a decrease in 6-carboxyfluorescein (6-FAM) at 520 nm in the *LNMAT1* (462 to 479 nt)/CCL2 TTS4 (-60 to -43 bp) group compared with that of the control ssRNA/ CCL2 TTS group (Fig. 5g-h), which was similar to the FENDRR/PITX2-positive control group (Fig. 5i). These data indicate that FRET took place between the fluorescein donor and the rhodamine acceptor, which directly correlates with the formation of triple helices³. Circular dichroism (CD) spectroscopy⁴ showed that the *LNMAT1* (462 to 479 nt)/CCL2 TTS4 (-60 to -43bp) group had a distinct peak at 270-280 nm and a strong negative peak at 210 nm (Fig. 5j-k), which was similar to the FENDRR/PITX2-positive control group (Fig. 5l), suggesting that *LNMAT1* directly formed triplexes with the CCL2 promoter sequence *in vitro*. The abovementioned results have been incorporated into the revised manuscript.

References:

- 1) Chu C, *et al.* Understanding RNA-Chromatin Interactions Using Chromatin Isolation by RNA Purification (ChIRP). *Methods Mol Biol***1480**, 115-123 (2016).
- 2) He S, *et al.* LongTarget: a tool to predict lncRNA DNA-binding motifs and binding sites via Hoogsteen base-pairing analysis. *Bioinformatics***31**, 178-186 (2015).
- 3) Reither S, *et al.* Specificity of DNA triple helix formation analyzed by a FRET assay. *BMC Biochem***3**, 27 (2002).
- 4) Mondal T, *et al.* MEG3 long noncoding RNA regulates the TGF-beta pathway genes through formation of RNA-DNA triplex structures. *Nature communications***6**, 7743 (2015).

Reviewer #2:

In the manuscript “Long noncoding RNA *LNMAT1* promotes lymphatic metastasis of bladder cancer via CCL2-dependent macrophage recruitment” the authors found and characterized a novel long noncoding RNA (*LMAT1*) implicated in lymphatic dissemination of bladder cancer cells. Additionally, the authors propose a mechanism where *LNMAT1* upregulates CCL2 in cancer cells leading to a higher recruitment of macrophages, which promotes lymphangiogenesis and metastasis. The results are interesting and novel, the mechanism of CCL2 activation by *LNMAT1* is well-characterized and the correlation with human data is remarkable. However, the manuscript seems incomplete, with a number of significant issues that need to be addressed, as described below.

Major comments

1: Materials and methods information about certain experiments (e.g. wound healing) is missing. Moreover, “Methods” and “Supplementary Materials and Methods” seem to be exactly the same, whereas those 2 parts should complement each other.

Response: We thank the reviewer for these comments. The Methods section in the main text was more detailed than the Supplementary Materials and Methods section in the originally submitted manuscript. As suggested by the reviewer, we have gone through both sections and made the appropriate corrections in the revised manuscript so that the sections complement each other. In addition, the descriptions for certain experiments, such as the wound healing assay and the cell isolation and differentiation assays, have been added to the revised manuscript.

2: The reviewer is not convinced about the cell isolation and differentiation protocols of human monocytes in macrophages. How is it possible to differentiate monocytes into macrophages without adding any differentiation factors (usually GM-CSF or M-CSF)? Furthermore, 24 hours of treatment with LPS or IL-4 are usually considered sufficient for macrophage polarization, so why did the authors decide to polarize them for 7 days? The authors also should mention the paper in which the protocol that they followed is reported. What kind of macrophages were used in experiments shown in Fig. 7 and Fig. 8? Why do the authors claim they are TAMs if they were not isolated from tumors?

Response: We appreciate the reviewer for raising these excellent concerns, and the

reviewer's comments are well taken.

(1) We apologize for the improperly provided protocols for cell isolation and differentiation and thank the reviewer for pointing this out. The original protocols for cell isolation and differentiation of human monocytes in macrophages used in our study were according to previously published reports^{1,2,3,4}. We are sorry for the inappropriate modifications that were made in the last version. The 7 days in the protocols is the total period of time needed for macrophage differentiation and polarization (6 days for differentiation plus 1 day for polarization). In brief, human monocytes were isolated from buffy coats prepared from healthy volunteer donors. Peripheral blood monocytes isolated by Ficoll-Hypaque for 50 min at 400g density gradient centrifugation were seeded at 2×10^6 /ml in 24-well plates in RPMI 1640 medium supplemented with 10% heatinactivated human AB serum, 50 U/ml of penicillin, 50 ug/ml of streptomycin 2 mM L-glutamine, supplemented with 100ng/ml human M-CSF to stimulate macrophage differentiation. Then 2×10^6 cells/mL monocyte were treated with differentiation factors (usually GM-CSF or M-CSF) for differentiation into macrophages for 6 days and further monocyte-derived macrophages were treated for 1 days with 25 μ g/ml lipopolysaccharide (LPS) in order to obtain M1-polarized macrophages or 45ng/ml recombinant human IL-4 for M2-polarized macrophages for *in vitro* activation. These cell isolation and differentiation protocols, with the appropriate modifications, have been added to the Methods section.

(2) We thank the reviewer for this comment and we are sorry that we did not clearly explain this point in our original manuscript. In Fig. 7 and Fig. 8, the macrophages treated with CM from *LNMT1*-transduced bladder cancer cells but not those treated with CM from control group displayed the stretched and elongated morphology and exhibited a CD206 high /HLA-DR low phenotype (Fig. 7e and Supplementary Fig. 11a), which was similar to the characteristics of TAMs^{1,4,5}, indicating the functional activation of macrophages. The abovementioned results have been incorporated into the revised manuscript.

References

- 1) Su S, *et al.* A positive feedback loop between mesenchymal-like cancer cells and macrophages is essential to breast cancer metastasis. *Cancer cell* **25**, 605-620 (2014).
- 2) Smith MP, *et al.* The immune microenvironment confers resistance to MAPK pathway inhibitors through macrophage-derived TNF α . *Cancer Discov* **4**, 1214-1229 (2014).
- 3) Sierra-Filardi E, *et al.* Activin A skews macrophage polarization by promoting a proinflammatory phenotype and inhibiting the acquisition of anti-inflammatory macrophage markers. *Blood* **117**, 5092-5101 (2011).

- 4) Chen J, *et al.* CCL18 from tumor-associated macrophages promotes breast cancer metastasis via PITPNM3. *Cancer cell* **19**, 541-555 (2011).
- 5) Mizukami Y, *et al.* CCL17 and CCL22 chemokines within tumor microenvironment are related to accumulation of Foxp3+ regulatory T cells in gastric cancer. *International journal of cancer* **122**, 2286-2293 (2008).

3: Could the authors assess the influence of corresponding treatments (Fig. 7C) on the expression of at least one more M2 marker and two chosen M1 markers (FACS)? For the assessment of capillary formation, could the authors use CM from TAMs sorted from murine tumors (overexpressing and silenced for *LNMAT1*)?

Response: We appreciate the reviewer for making these excellent suggestions and the reviewer's points are well taken. As requested by the reviewer, we assessed the expression of specific M2 marker (CD206 and CD163) and M1 markers (HLA-DR and CD86) using FACs. As shown in Fig. 7e and Supplementary Fig. 11b-c, treatment of CM derived from *LNMAT1*/cells significant increased the expression of the specific M2 marker (CD206 and CD163) but decreased the expression of M1 markers (HLA-DR and CD86), compared to vector-control cells, which provided more evidence that *LNMAT1* overexpression in cancer cells induces macrophage activation.

As requested by the reviewer, we further examined the effect of TAMs sorted from murine tumors on capillary formation. As shown in Supplementary Fig. 12a-b, treatment with CM derived from TAMs sorted from murine *LNMAT1*/tumors significantly increased lymphatic capillary formation compared with CM derived from TAMs sorted from murine vector/tumors, which further supports the notion that *LNMAT1* overexpression in cancer cells promotes lymphangiogenesis. The abovementioned results have been incorporated into the revised manuscript.

4: Except for microlymphatic vessel density, could the authors also assess the presence of big LVs in murine tumors sh-NC vs sh-*LNMAT1* (Fig. 2H,I), as usually such bigger vessels participate in lymphatic metastasis? Moreover, the authors quantified signal of "LYVE-1 positive cells", and attributed it to lymphatic vessels. How do the authors know that they are not macrophages? Could the authors use double fluorescence staining (F4/80 and LYVE-1 or F4/80 and VEGFR3) in order to quantify LVs exclusively?

Response: We thank the reviewer for raising these excellent suggestions and the reviewer's

comments are greatly appreciated. Through the reassessment of IHC staining in the vector/tumors and *LNMAT1*/tumors, we did observe significant decreased big LVs in murine sh-*LNMAT1*/tumors compared with sh-NC/tumors in both the intratumoral and peritumoral regions ($p < 0.05$; $p < 0.05$) (Supplementary Fig. 5a-b). Moreover, as suggested by the reviewer, the previous improperly prepared images presented with small in LVs have been replaced by images presented with big LVs in the revised manuscript.

As requested by the reviewer, we further examined macrophages with an anti-F4/80 antibody and lymphatic vessels with an anti-LYVE-1 antibody through double staining assays. As shown in Supplementary Fig. 5d, the F4/80⁺-cells were distinct with LYVE-1⁺-cells, which is consistent with previous reports, which LYVE-1 is considered as a common lymphatic marker that is extensively used for the detection of lymphatic vessels^{1,2}. The abovementioned results have been incorporated into the revised manuscript.

References

- 1) He W, *et al.* Long noncoding RNA BLACAT2 promotes bladder cancer-associated lymphangiogenesis and lymphatic metastasis. *J Clin Invest* **128**, 861-875 (2018).
- 2) Liu L, *et al.* TBL1XR1 promotes lymphangiogenesis and lymphatic metastasis in esophageal squamous cell carcinoma. *Gut* **64**, 26-36 (2015).

5: The authors demonstrate the presence of increased metastasis upon *LNMAT1* overexpression in both tail vein and tumor inoculation experiments. Thus, does it observation relay more on increased tumoral lymphangiogenesis or increased extravasation of cancer cells? Could the authors assess the metastasis levels from primary tumors upon the blockage of lymphangiogenesis (e.g. with anti-VEGF-C)? Moreover, could the authors assess the migration of cancer cells (with and without *LNMAT1* overexpression) on the invasion through lymphatic endothelium monolayer *in vitro*?

Response: We appreciate the reviewer for raising these excellent suggestions and the reviewer's comments are greatly appreciated. As requested by the reviewer, the effect of VEGF-C inhibition by the anti-VEGF-C antibody on *LNMAT1*-induced LN metastasis was examined *in vivo*. As shown in Fig. 8f-g, blockade of the VEGF-C signaling using a VEGF-C neutralizing antibody (pV1006R-r) significantly decreased the intratumoral and peritumoral lymphatic vessel densities and reduced the lymphatic metastasis capability of *LNMAT1*-overexpressing cells, suggesting that blocking of VEGF-C signaling abrogated the *LNMAT1*-induced lymphangiogenesis and lymphatic metastasis *in vivo*.

We further examined the effect of *LNMAT1* overexpression on the invasion of cancer cells through a lymphatic endothelium monolayer *in vitro*. As shown in Supplementary Fig. 12c, the overexpression of *LNMAT1* significantly promoted the migratory and invasive capability of bladder cancer cells through the lymphatic endothelium monolayer, suggesting that *LNMAT1* may also increase the invasiveness of cancer cells. It has been previously reported that there is a contributing role for CCL2 in cancer cell metastasis^{1,2}. Here, we demonstrated that *LNMAT1* epigenetically activated CCL2 expression by recruiting hnRNPL to the CCL2 promoter. Therefore, *LNMAT1*-induced CCL2 might play dual roles in the lymphatic metastasis of bladder cancer cells. These abovementioned results have been incorporated into the revised manuscript.

References

- 1) Zhang J, *et al.* CC chemokine ligand 2 (CCL2) promotes prostate cancer tumorigenesis and metastasis. *Cytokine Growth Factor Rev.* **21**(1):41-8 (2010).
- 2) Nam JS, *et al.* Chemokine (C-C motif) ligand 2 mediates the prometastatic effect of dysadherin in human breast cancer cells. *Cancer Res.* **66**(14):7176-84 (2006).

Minor comments:

1: The manuscript has a many grammar mistakes that need to be corrected (e.g. “lncRNAs regulate chemokine activation by interact with chromatin” should be “lncRNAs regulate chemokine activation by interaction with chromatin”). Please revisit the whole manuscript.

Response: We thank the reviewer for these comments, which are well taken. We have carefully edited the entire manuscript and have had the revised manuscript corrected again by professional editors before this resubmission.

2: Could the authors indicate what was the LN status in high-grade muscle invasive bladder cancers (MIBC) used for NGS analysis (Fig. 1A).

Response: We thank the reviewer for this comment. Five high-grade muscle invasive bladder cancer tissues (MIBC) were used for NGS analysis in Fig. 1A, including three LN-positive bladder cancer tissues and two LN-negative bladder cancer tissues. As suggested by the reviewer, the characteristics of these MIBC patients used in the NGS analysis have been added to the revised manuscript and Supplementary Table 1.

3: Could the authors assess the effect of CCL2 blockage *in vitro* (a migration or a wound healing assay with cancer cells)?

Response: We thank the reviewer for this comment and the reviewer's point is well taken. As suggested by the reviewer, we further examined the effect of CCL2 inhibition *in vitro* via a migration assay with cancer cells. Consistent with previous reports^{1,2}, CCL2 inhibition significantly inhibited the migratory capability of bladder cancer cells (Supplementary Fig. 6a-b). The abovementioned results have been incorporated into the revised manuscript.

References

- 1) Zhang J, *et al.* CC chemokine ligand 2 (CCL2) promotes prostate cancer tumorigenesis and metastasis. *Cytokine Growth Factor Rev.* **21**(1):41-8 (2010).
- 2) Nam JS, *et al.* Chemokine (C-C motif) ligand 2 mediates the prometastatic effect of dysadherin in human breast cancer cells. *Cancer Res.* **66**(14):7176-84 (2006).

4: In Fig. 6G, H the cells lines were silenced for hnRNPLs, however in the text referring to this figure the authors state that the cells were transduced to express hnRNPLs. Which version is correct?

Response: We appreciate the reviewer's comment, and we thank the reviewer for pointing out this error. It should be "silencing" as Figure indicated. We have gone through the whole manuscript and made the appropriate corrections, including revising the descriptions for Fig. 6G-H.

5: What are the p values in Fig. 2G and Fig. 4J?

Response: We thank the reviewer for this question. *P*-values have been added to Supplementary Fig. 3a-b, which was Fig. 2G in the original manuscript, and Supplementary Fig. 7c, which was Fig. 4J in the original version.

6: As mentioned in Figure 1, three lncRNAs, (*LNMAT1*, CTD-2231H16.1 and BCAR4) were consistently upregulated both in the MIBC and LN-positive bladder cancer tissues (Figure 1C). Beside the significant overexpression of *LNMAT1* in figure 1E, could the authors also display the expression levels of the other 2 lncRNAs in the Supplementary figures?

Response: We thank the reviewer for this comment and the reviewer's point is well taken. As requested by the reviewer, the expression levels of BCAR4 and CTD-2231H16.1 have been incorporated into Supplementary Fig. 1b-c in the revised manuscript.

7: The authors need to correct figure legend accuracy (e.g. no figure legend in Fig. 3G; Fig. 3D appears twice in the legend; what do the arrows in Fig. 4I indicate?).

Response: We appreciate the reviewer's comments, and we thank the reviewer for pointing out the mistakes. All the figure legends have been carefully edited and appropriate corrections, including those for Fig. 3D and 3G, have been made in the revised manuscript.

We apologize that we did not explain what the arrows indicated in Fig.4I in our original manuscript. The arrows in Fig. 4I indicate the magnified inset area. To avoid confusion, we have changed the arrows to the red box, which is explained in the figure legend.

8: Some figures do not appear in the order introduced in the text.

Response: We apologize for the mistakes and thank the reviewer for pointing these out. We have gone through the entire manuscript and have made appropriate corrections into the revised manuscript.

Reviewer #3:

The manuscript by Chen et al. focused on the role of a long non-coding RNA *LNMAT1* in lymphnode metastasis from bladder cancer. This study is an overall interesting story. The major novel contribution is the role of *LNMAT1* in lymphatic infiltration by bladder cancer cells, and the mechanism through which *LNMAT1* regulates *CCL2*. There are a lot of data ranging from human clinical specimens to mouse models to molecular mechanisms. The mechanistic studies, particularly the regulation of *CCL2* expression, were well done, and convincing to me. The correlation between *CCL2* and *LNMAT1* in human cancer data is also supporting the mechanism and very nice. However, a few major concerns on the *in vivo* experiments exist that need to be properly addressed.

Major comments

1: In the *in vivo* model, the authors showed that modulating *LNMAT1* leads to changes in lymph node size and lymphatic infiltration. However, they did not show any data on the primary tumor size. An equally possible alternative explanation of their data is that *LNMAT1* regulates primary tumor size, and larger primary tumor leads to increased lymphatic infiltration. Judging from the luciferase images (e.g. fig 4i), the primary tumor size seems to correlate with *LNMAT1* modulation.

Response: The reviewer's comment is well taken. We appreciate the reviewer for pointing out the improperly prepared Fig. 4i and we are sorry that we did not explain this clearly in our original manuscript. The impact of *LNMAT1* on lymphatic metastasis was checked when the vector-control tumors (approximately 32 days) reached the same size as the *LNMAT1*/tumor (approximately 26 days). However, the images presented in the Fig. 4i in the original manuscript showed different tumor size. As suggested by the reviewer, the appropriate image has been changed in the Fig. 4i and the times related to the vector-control tumors (approximately 32 days) and *LNMAT1*/tumor (approximately 26 days) have been added to the revised manuscript.

2: The critical experiment in which anti-*CCL2* antibody treatment reduces lymphatic infiltration lacks key controls. This experiment is key to pinpoint a functional role of tumor-cell-derived *CCL2* in the lymphatic infiltration process. Currently, anti-*CCL2* is only used in mice with cells overexpressing *LNMAT1*. But it is equally possible that anti-*CCL2* is inhibiting *CCL2* from other non-cancer cell types *in vivo*—if true, one would predict that

anti-CCL2 would be effective in control bladder cancer cells as well.

Response: We appreciate the reviewer's comments and the reviewer raises a very interesting and important question. After submission of our manuscript, we realized that our data were not sufficient to support our conclusion, which the tumor-cell-derived CCL2 was the key mediator for *LNMAT1*-induced lymphatic metastasis. Therefore, 2 sets of experiments were immediately started. First, we examined the inhibitory effect of CCL2 via anti-CCL2 antibody on the lymphatic metastasis of control bladder cancer cells, which was also suggested by the reviewer. The anti-CCL2 antibody only partially inhibited the LN-metastatic capability of the control bladder cancer cells (the ratio of metastatic LNs from 37.50% reduced to 25.00%) but strongly inhibited the LN-metastatic capability of *LNMAT1*-overexpressing cells (the ratio of metastatic LNs from 93.75% reduced to 18.75%) (Supplementary Fig. 7c), suggesting that *LNMAT1*-induced CCL2 in tumor cells played a vital role in *LNMAT1*-induced lymphatic metastasis. Second, the effect of tumor-cell-derived CCL2 on *LNMAT1*-induced lymphatic metastasis was further examined via knockdown of CCL2 expression in vector and *LNMAT1*-overexpressing cells. Consistently, silencing CCL2 also only moderately reduced the LN-metastatic capability of vector-control bladder cancer cells (from 31.25% reduced to 18.75%) but remarkably inhibited the LN-metastatic capability of *LNMAT1*-overexpressing cells (from 81.25% to 12.50%; Fig. 4l-m and Supplementary Fig. 7d). Taken together, these results provided further evidence that tumor cell derived CCL2 is a key mediator for *LNMAT1*-induced lymphatic metastasis. These abovementioned results have been incorporated into the revised manuscript.

3: The *in vivo* tumor model has limitations—it is human bladder cells injected in a non-physiological site (foot pad), with the lack of effective adaptive immunity (using nude mice). I think the authors should at least discuss these limitations.

Response: We thank the reviewer for these comments and the reviewer's points are well taken. In this study, popliteal lymph node metastasis model was employed by injecting cells into mouse foot pads. Although it has been reported that footpad injection is a sensitive and quantitative way to measure lymphatic metastasis *in vivo*^{1,2,3}, multiple limitations of this model need to be noticed. First, the microenvironment of the footpads is quite different from the bladder microenvironment, and thus the mechanism in which the formation of microlymphatic vessels within both the bladder cancer and bladder cancer-adjacent tissues in

footpads is different than in the bladder. Moreover, the intratumoral interstitial fluid pressure (IFP) is commonly much higher than that in the surrounding host tissues, which influences the different lymphatic fluid reflux and filtration rates, causing some differences in the results of lymphatic metastasis⁴. Thus, it would be better to inoculate *LNMAT1*^{+/-}UMUC-3 cells or tumors formed by *LNMAT1*^{+/-}UMUC-3 cells directly into the bladder walls of mice to examine the effect of *LNMAT1* induced lymph node metastasis. In addition, the BALB/c nude mouse model lacks effective adaptive immunity and is commonly used in studies to test the pharmacological treatment of human tumor xenografts⁵. Therefore, the effect of *LNMAT1* induced-lymph node metastasis could also be analyzed in mouse cell lines using BALB/c mice with human immunity. The abovementioned descriptions have been added to the Discussion section in the revised manuscript.

References

- 1) Ito T, *et al.* Pituitary tumor-transforming 1 increases cell motility and promotes lymph node metastasis in esophageal squamous cell carcinoma. *Cancer research***68**, 3214-3224 (2008).
- 2) He W, *et al.* Long noncoding RNA BLACAT2 promotes bladder cancer-associated lymphangiogenesis and lymphatic metastasis. *J Clin Invest***128**, 861-875 (2018).
- 3) Liu L, *et al.* TBL1XR1 promotes lymphangiogenesis and lymphatic metastasis in esophageal squamous cell carcinoma. *Gut***64**, 26-36 (2015).
- 4) Avery M, *et al.* Lymph flow from murine footpad tumors before and after sublethal hyperthermia. *Radiat Res***132**, 50-53 (1992).
- 5) Shultz LD, *et al.* Humanized mice in translational biomedical research. *Nat Rev Immunol***7**, 118-130 (2007).

Minor comments:

1: The authors claim the use of “*in vitro* synthesized hnRNPL”, but I couldn’t find this info in the methods. Is this truly “synthesized”, or it is purified recombinant protein expressed in bacteria or another host?

Response: We apologize for the error and thank the reviewer for pointing this out. The “*in vitro* synthesized hnRNPL” is purified recombinant hnRNPL protein expressed in bacteria. Appropriate corrections have been made in the revised manuscript, where “*in vitro* synthesized hnRNPL” was replaced by “purified recombinant hnRNPL” (Fig. 5o and Fig. 6c).

2: The histology pictures (e.g. Fig 2h, 2i, 7a, 7b) are too small with low resolution of details. Not very easy to see even after expanding on computer screen. Maybe including the large versions in supplement?

Response: We thank the reviewer for the comment, and the reviewer's point is well taken. As suggested by the reviewer, all the histology pictures (Fig.2h, 2i, 7a and 7b) have been replaced with high-resolution images, and larger version have been added to the revised manuscript (Fig. 7a-d and Supplementary Fig. 4a-b).

REVIEWERS' COMMENTS:

Reviewer #1 (Remarks to the Author):

It would appear that the authors have addressed the large majority of comments of each reviewer and that the manuscript is improved considerably. I have no further suggestions.

The figures may still be a bit small but I think this can be addressed by the editorial process in production

Reviewer #2 (Remarks to the Author):

The manuscript has been extensively improved. Below the remaining issues:

1)The reviewer appreciates the evaluation of additional M1 and M2 markers following corresponding treatments (Fig. 7E, Supp Fig. 11 C). Nevertheless, statistical analysis is missing.

2)The authors should mention in Results and in Material and Methods that LNMAT1 influences the tumor growth and that lymphatic metastasis were checked when the vector-control tumors reached the same size as the LNMAT1 tumors. This is an important information that is totally missing in all the manuscript.

3) Human derived macrophages in Fig. 7 and 8 should be called TAM-like macrophages, and not TAMs.

Reviewer #4 (Remarks to the Author):

the authors have addressed all the concerns of referee 3 and subsequent modifications have been inserted in the figures and text adequately. The manuscript is now good for publication.

Response to Reviewers' comments and suggestions:

Reviewer #1:

It would appear that the authors have addressed the large majority of comments of each reviewer and that the manuscript is improved considerably. I have no further suggestions. The figures may still be a bit small but I think this can be addressed by the editorial process in production.

Response: We deeply thank the reviewer for the appreciation on our tremendous efforts in addressing all the concerns.

Reviewer #2:

The manuscript has been extensively improved. Below the remaining issues:

1. The reviewer appreciates the evaluation of additional M1 and M2 markers following corresponding treatments (Fig. 7E, Supp Fig. 11 C). Nevertheless, statistical analysis is missing.

Response: We thank for the reviewer's comment and the reviewer's point is well taken. As suggested by the reviewer, the statistics has been added to Fig.7C, which was Fig. 7E in the original manuscript, and Supplementary Fig.11C in the revised manuscript.

2. The authors should mention in Results and in Material and Methods that LNMAT1 influences the tumor growth and that lymphatic metastasis were checked when the vector-control tumors reached the same size as the LNMAT1 tumors. This is an important information that is totally missing in all the manuscript.

Response: We thank for the reviewer's comment and the reviewer's point is well taken. As suggested by the reviewer, appropriate sentences have been added in the Results and Methods sections in the revised manuscript. Please refer to lines 1-2, Page 8 and lines 21-22, Page 26.

3. Human derived macrophages in Fig. 7 and 8 should be called TAM-like macrophages, and not TAMs.

Response: We thank for the reviewer's comment and the reviewer's point is well taken. Appropriate modification has been made in the Figures 7 and 8 in the revised manuscript.

Reviewer #4:

The authors have addressed all the concerns of referee 3 and subsequent modifications have been inserted in the figures and text adequately. The manuscript is now good for publication.

Response: We thank the reviewer for the positive comment on our revised manuscript.